# Seasonal influence of snow conditions on Dall's sheep productivity in Wrangell-St Elias National Park and Preserve

Christopher L. Cosgrove[1]*, Jeff Wells[2], Anne W. Nolin[1,3], Judy Putera[4], Laura R. Prugh[5]

1 College of Earth Ocean and Atmospheric Sciences, Oregon State University, Corvallis, OR, United States of America, 2 Alaska Department of Fish and Game, Tok, AK, United States of America, 3 Department of Geography, University of Nevada Reno, Reno, NV, United States of America, 4 Wrangell-St. Elias National Park and Preserve and Central Alaska Inventory & Monitoring Network, AK, United States of America, 5 School of Environmental and Forest Sciences, University of Washington, Seattle, WA, United States of America

* cosgrovc@oregonstate.edu

**Data Availability Statement:** Daily rasters of snow properties, generated by SnowModel as described in the manuscript, will be held at the Oak Ridge National Laboratory (ORNL) DAAC, https://daac.

## Abstract

Dall's sheep (*Ovis dalli dalli*) are endemic to alpine areas of sub-Arctic and Arctic northwest America and are an ungulate species of high economic and cultural importance. Populations have historically experienced large fluctuations in size, and studies have linked population declines to decreased productivity as a consequence of late-spring snow cover. However, it is not known how the seasonality of snow accumulation and characteristics such as depth and density may affect Dall's sheep productivity. We examined relationships between snow and climate conditions and summer lamb production in Wrangell-St Elias National Park and Preserve, Alaska over a 37-year study period. To produce covariates pertaining to the quality of the snowpack, a spatially-explicit snow evolution model was forced with meteorological data from a gridded climate re-analysis from 1980 to 2017 and calibrated with ground-based snow surveys and validated by snow depth data from remote cameras. The best calibrated model produced an RMSE of 0.08 m (bias 0.06 m) for snow depth compared to the remote camera data. Observed lamb-to-ewe ratios from 19 summers of survey data were regressed against seasonally aggregated modelled snow and climate properties from the preceding snow season. We found that a multiple regression model of fall snow depth and fall air temperature explained 41% of the variance in lamb-to-ewe ratios ($R^2 = .41$, $F(2,38) = 14.89$, $p<0.001$), with decreased lamb production following deep snow conditions and colder fall temperatures. Our results suggest the early establishment and persistence of challenging snow conditions is more important than snow conditions immediately prior to and during lambing. These findings may help wildlife managers to better anticipate Dall's sheep recruitment dynamics.

## Introduction

The terrestrial ecology of the Arctic Boreal region (ABR) is changing rapidly as a result of amplified increases in temperatures [1–4]. Seasonal snow coverage exists in the ABR for up to

ornl.gov. Scripts that process this daily data into the seasonal aggregates used in the analysis will be additionally included. This data is not possible to submit as supplementary information given its multiple terabyte size and is in the process of being archived. Field data used to calibrate SnowModel, as described in the manuscript, is already archived at the ORNL DAAC - see https://daac.ornl.gov/cgi-bin/dsviewer.pl?ds_id=1656. Scripts and methods used in the calibration of SnowModel to these data are included in the Supplementary Information. Dall's sheep survey data and the seasonally aggregated, SnowModel-derived, snow property data are included in the Supplementary Information alongside scripts preparing them for analysis (S2 to S10). All scripts used to generate figures are included in the Supplementary Information. Additional data used to create figure A2 in S1 Appendix can be located at https://daac.ornl.gov/ABOVE/guides/Last_Day_Spring_Snow.html.

**Funding:** AWN and CLC received funding from grant number NNX15AV86A from the National Aeronautics and Space Administration Terrestrial Ecology Program - https://above.nasa.gov/cgi-bin/inv_pgp.pl?pgid=3379&projType=project&projID=3379&progID=6. LRP received funding from grant number NNX15AU21A from the National Aeronautics and Space Administration Terrestrial Ecology Program - https://above.nasa.gov/cgi-bin/inv_pgp.pl?pgid=3379&projType=project&projID=3379&progID=6. The funders had no role in study design, data collection and analysis, decision to publish, or preparation of the manuscript.

**Competing interests:** The authors have declared that no competing interests exist.

10 months annually and profoundly impacts ecosystem function. Studies point towards continued reduction in the annual duration of snow cover and overall accumulation in the ABR, with region and elevation dependent variations in trend and severity [5]. Mid-winter warming events have been seen to cause substantial alteration to snow properties and the incidence and severity of these events are thought to be increasing [6–8]. Snow processes have been linked to the population dynamics, movement, habitat selection, and life-cycles of a wide variety of mammals living in the ABR ranging in size from polar bears (*Ursus maritimus*, [9]) and moose (*Alces alces*, [10]), through to lemmings (*Lemus lemus*, [11]) and snowshoe hares (*Lepus americanus*, [12]). Due to their importance to Northern societies, ungulates native to the ABR, such as moose, caribou (*Rangifer tarandus*) and muskoxen (*Ovibos moschatus*) have been subject to broad scientific enquiry [10, 13–20]. These studies indicate that ungulate populations in the ABR are negatively affected by extreme conditions that could increase in severity and frequency due to climate change [21, 22]. For example, 'locked-pastures', where access to winter forage is restricted through either deep snow or ice-layers, have been linked to caribou and muskox mass mortality events [22–25].

Snow cover in mountain areas is highly variable in both space and time [26] as the interplay of temperature, precipitation, solar radiation, vegetation cover and wind produces intricate patterns of depth, density and stratigraphy in complex terrain. While remote sensing products utilising optical and infrared wavelengths have some ability to detect this variability, their coarse spatial grain (~500 m) at daily time scales, impediment by cloud cover, and inability to quantify snow depth and density, limit their application in snow ecology questions [27]. Passive microwave derived remote-sensing products have shown promise in mapping snow properties such as water equivalent [28] and rain-on-snow events [29], but these products currently have a spatial resolution of >5 km, limiting their usefulness in mountain contexts.

Physically-based snow evolution models offer a promising means of obtaining a variety of snow properties that cannot be obtained from remote sensing alone. These models solve the surface mass-energy balance to map snow properties at a user-defined spatial and temporal resolution. However, there has been limited application of these models in wildlife research relative to those incorporating remotely sensed snow data, possibly due to the different technical skills required. Models have been used to simulate detailed snow data at single point locations for comparison to long-term wildlife data [24, 30], or to quasi-spatialize a single grid cell model at a coarse, 45 km resolution [31]. To our knowledge, no study has yet exploited the ability of modern snow models to produce longer time series of spatially-distributed data to compare to population dynamics of wildlife. Here, we use a leading snow evolution model, SnowModel [32], capable of operating with a 3D snow redistribution sub-model [33], to map daily snow and climate conditions at a high spatial resolution for a mountainous sub-Arctic domain inhabited by a population of Dall's sheep (*Ovis dalli dalli*) that has been surveyed periodically over the past 50 years. The advantage of this approach is that it allows identification of important seasonal snow properties, and allows the simulation of snow conditions across Dall's sheep alpine habitat as opposed to potentially non-representative point-locations, such as meteorological stations in valley-bottoms [34].

We examined the importance of the preceding season's snow conditions on summer lamb production of Dall's sheep in Wrangell-St Elias National Park and Preserve, Alaska, USA (WRST) using model derived covariates of snowpack quality. Dall's sheep are a wild ungulate endemic to mountains of the ABR in north-western North America and are an important herbivore in high-latitude alpine ecosystems that may be acutely vulnerable to climate change [35]. They are also a highly prized Alaskan game species [41]. Dall's sheep often use windward aspects during snow-covered months, where they rely on wind-scoured patches of snow-free or soft and shallow snow-covered forage to buffer caloric deficit [37]. Populations of Dall's

sheep have historically fluctuated widely in size [36, 38–40]. These fluctuations are thought to be largely governed by variations in the production and survival of lambs, as adult survival has been shown to be relatively stable except after extreme winter events [41, 42], and only a limited number of mature rams are harvested each year [43]. Mature Dall's sheep ewes typically produce one lamb in mid-May to early-June [44], and decreased summer production and survival of lambs has been linked to adverse winter weather and persistent or deep snow conditions [38, 42, 45–47]. However, previous studies have relied upon remotely-sensed snow cover phenology metrics, with vertical properties of snow, e.g. greater depth and density, inferred from the longer persistence of snow covered areas [42, 47]. Thus, the seasonal importance of different snow properties such as depth and density on Dall's sheep remains unknown.

Snow properties are thought to affect ungulates such as Dall's sheep in 3 main ways. First, access to forage may be restricted where snow is deeper or harder [48]. Second, movement may be energetically expensive where deeper snow does not support an animal's weight [49]. Third, susceptibility to predation may be enhanced in deep snow conditions where the snow density supports a predator's foot load but impedes movement of an ungulate [50]. Forage restriction from deep or hard snow cover established in fall has been shown to have stronger impacts on reindeer populations than restriction later in the winter or spring [51], but whether these patterns occur in mountainous regions with more heterogeneous snow properties is not known.

Here, we examine the relationships between preceding snow conditions and Dall's sheep productivity, measured as the number of lambs per ewe-like sheep (hereafter, lamb-to-ewe ratios). Our methodology affords the novelty of examining *when* and *which* snow properties are most important. In other studies of alpine ungulates and Dall's sheep low winter temperatures and high snowfall have been shown to decrease summer productivity [e.g., 45, 52], so we study these climate variables for influence relative to, and in combination with, model derived snow properties. Additionally, we present trends in modelled snow and climate covariates from 1980 to 2017 to shed light on potential linkages between climate change, snow properties, and Dall's sheep population dynamics.

To establish the relative importance of the seasonality of snow conditions we tested two contrasting hypotheses: (H1) the cumulative effects of persistent snow conditions that are unfavourable for Dall's sheep productivity will be most important, in which case snow conditions established in the fall months and persisting through the winter months should better explain summer lamb-to-ewe ratios; (H2) snow conditions in the lambing season will have the strongest effect, in which case snow conditions in the spring months should better explain lamb-to-ewe ratios. As adult survival is considered stable relative to that of Dall's sheep lambs, our first hypothesis proposes that the effect of snow conditions indirectly influences lamb production and survival via ewe body condition, which is affected by the winter-long accumulative effect of snow conditions aiding or abetting forage and movement. The second hypothesis instead emphasises that snow conditions may have a more direct influence on lamb survival, and hence productivity, both through their effect on foraging and movement immediate to and after birth.

## Materials and methods

### Study area

Our study area was a 8,678 km$^2$ region located in northern Wrangell-St Elias National Park and Preserve (WRST; 62˚18'46"N, 143˚ 15' 31"W; Fig 1). A small portion of the study area was outside WRST and included portions of state, U.S. Fish and Wildlife Service, and private lands. Our study area falls within the Southeast Interior Alaska climate division, as mapped by Bieniek et al. [53]. Precipitation is relatively low, given the rain-shadowing of the Chugach mountain range to the south, and falls predominantly in May through to October. The annual

range of mean monthly temperatures is ~15˚C in July to ~-20˚C in January [53]. In the subalpine zone (1200–1400 m.a.s.l), patches of 1 to 2 m high dwarf birch (*Betula glandulosa*) and willow (*Salix* spp.) are separated by lichens and moss [54]. Alpine areas (> 1400 m.a.s.l) are either dry communities of low, matted alpine vegetation, consisting mostly of *Dryas*, or moist areas of grasses (*Festuca* spp. and *Poa* spp.) and sedges (*Carex* spp.) with occasional patches of low willow and birch shrubs [54]. Dall's sheep habitat extends from shrubline (~1400 m) into alpine areas where they favor areas close to rugged escape terrain [55]. Using Moderate Resolution Imaging Spectroradiometer (MODIS) derived snow cover data from 2000 to 2015, Cherry et al. (2017) found a median start of the continuous snow season (CSS) of the 26th September (±32 days SD) for elevations between 1219 m and 1524 m, and 30th August (±34 days SD) for elevations above 1524 m, across Denali National Park, Yukon Charley National Preserve and WRST. The median date for the end of the CSS at these elevations were respectively 30th May (±37 days SD) and the 28th June (±34 days SD) [56].

## Survey unit selection

Within WRST there are 34 survey units in which summer Dall's sheep surveys are conducted by the Alaska Department of Fish and Game (ADF&G) and National Park Service (NPS) (Fig 1). These are delineated by high elevation terrain bounded by water courses or glaciated valleys and are kept to a manageable size for surveying. We used survey data from 9 northern units that were selected based on proximity and similarity to the Jacksina survey unit (JSU) where our ground-based snow surveys were conducted (Fig 1). In the absence of long-term in-situ snow cover data within each survey unit, we used a 500 m MODIS-based remote sensing product, snow disappearance date (SDD), to identify units with similar snow cover phenology as the JSU from 2000–2016 [60]. We evaluated all units whose center point was within 100 km of the centre of the JSU (n = 17 units; Fig 1 in S1 Appendix). This search diameter of 200 km approximates to the maximum meso-β scale length forwarded by Orlanski [61] as typical for mountain disturbances on meteorology, thus ensuring all units had similar climatic influences. SDDs were generally later for units south of the JSU, whereas units to the north, east, and west had similar values (Fig 2 in S1 Appendix), suggesting the high-elevation ice-fields that separated the northern and southern units influenced snow conditions. Thus, we used sheep survey data from units 1 (Mentasta Mountains), 2 (Mount Sandford), 4E (Cross Creek), 4W (Nikonda Creek), 5E (Mount Allen), 5W (Stone Creek), 7W (Chisana) and 10 (Mount Drum), alongside that of the JSU, unit 3 (Fig 1).

## Sheep surveys

Sheep survey data was obtained from a collated dataset of state and federal monitoring surveys conducted by ADF&G and NPS. A study period of 1980 to 2017 was determined by the availability of meteorological forcing data for SnowModel (see below), and within this period 19 years of sheep survey data were available from 41 surveys in our selected survey units (Table 1 in S1 Appendix, Fig 1). The earliest survey date was 21st June and the latest the 4th of August. Mean lamb-to-ewe ratio was 0.30 (Max. = 0.55; Min. = 0.09, SD ±0.10) and the mean total sheep counted in each survey was 654 (Max. = 2549 Min. = 87, SD ±564). Surveys were conducted using either a small fixed-wing plane or by helicopter and all followed a minimum count method [62]. We note that aerial minimum count methods are subject to potential biases in comparison to distance-based population estimates [63] but we only use full surveys, i.e. where the entire Survey Unit is reported as covered, in our dataset. The difficulty of distinguishing the sex of non-mature Dall's sheep via aerial survey can lead to yearlings of both sexes and small-horned rams often being counted as ewes. A 'ewe-like' category is often used due to

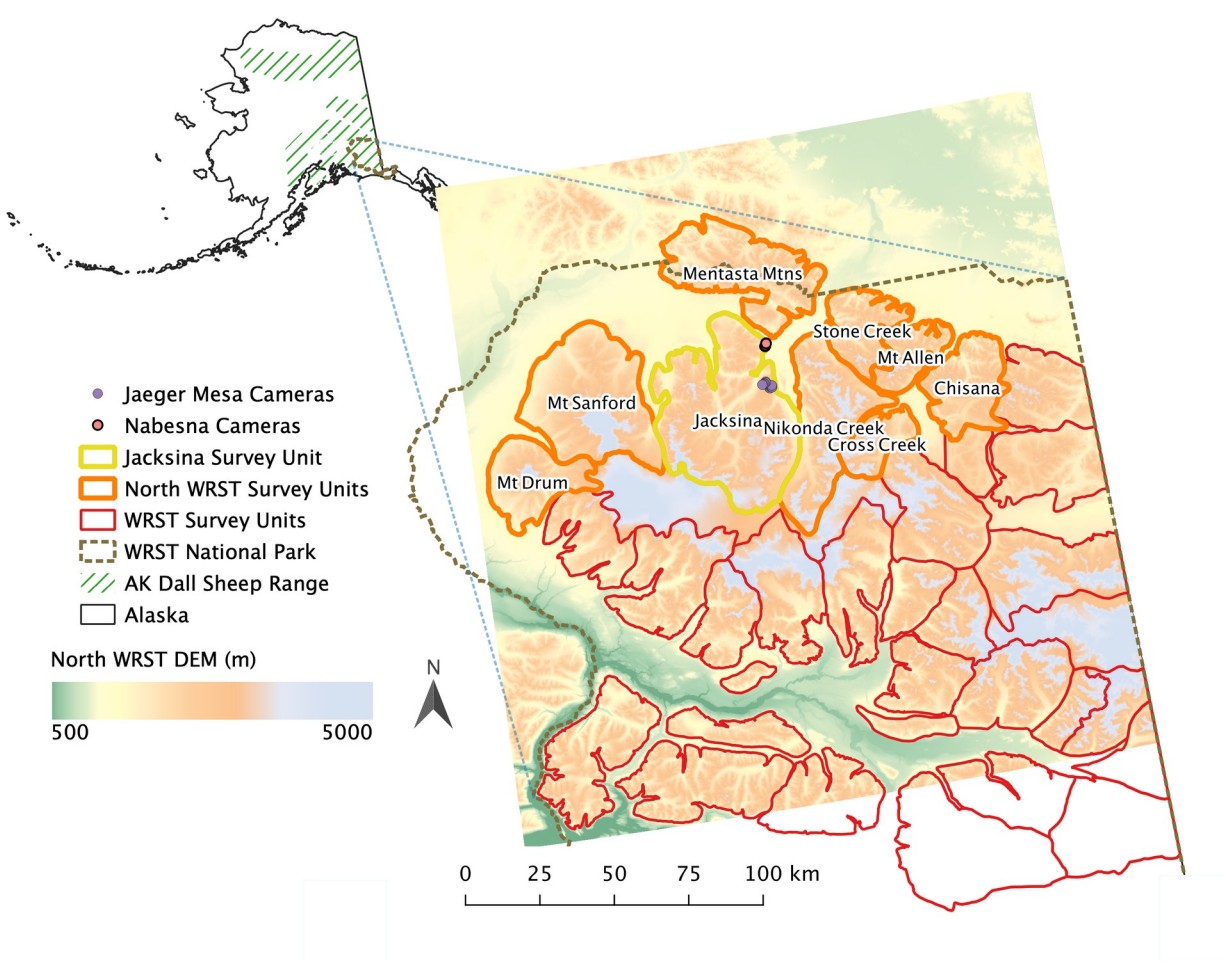

**Fig 1. Map of study area located in the northern Wrangell-St. Elias National Park and Preserve (WRST; brown dashed outline) and Alaska (inset).** Field-based snow surveys, including the installation of remote cameras upon Jaeger Mesa and near Nabesna, took place in the central Jacksina sheep survey unit (yellow outline) to calibrate a physically-based, spatially distributed snow evolution model. With the calibrated model we simulated daily snow conditions for high-elevation Dall's sheep terrain within the Jacksina survey unit domain from 1980 to 2017. A remote sensing analysis determined that the mean snow disappearance date (SDD) in 8 other survey units (outlined orange) was more similar to that of Jacksina compared to that of other units in the WRST (outlined red). We hence used observations of summer lamb-to-ewe ratios from Jacksina and these 8 nearby units to compare to model-derived metrics of seasonal snow conditions. GIS data for sheep survey units and WRST park boundary were sourced from [57, 58] respectively, the background digital elevation model is built from 1 Arc-second Digital Elevation Models (DEMs) of the United States Geological Service National Map 3D Elevation Program [59].

this uncertainty, and we therefore used reported 'ewe-like' counts as the denominator in lamb-to-ewe ratios where they are available. While this ratio is not a perfect measure of productivity because it is affected by a combination of factors including parturition rates, lamb survival, and adult survival, the juvenile-to-female ratios have been shown to be a useful measure of productivity in other ungulate species because the majority (96%) of the variation in the ratio is caused by variation in juvenile survival [64]. The inclusion of 'ewe-likes' leads to lower values than the true lamb-to-ewe ratio, but it is still a useful index of productivity and has been used as such in other Dall's sheep studies [56, 57].

## SnowModel

Snow and climate covariates were produced using SnowModel [32] at a daily timestep for the Jacksina study domain. SnowModel has been used successfully in wide variety of latitudinal

settings and has previously been used for studies in continental Alaska and mountain regions [64–66] SnowModel's five sub-models, MicroMet [67], EnBal [68], SnowPack [69], Snow-Tran-3D [33], and SnowAssim [70] in combination with topographic, land cover and meteorological data simulate a comprehensive set of snowpack evolution processes in a physically based manner (please refer to sub-model references for details on their physics and validation). MicroMet ingests meteorological data and distributes them throughout the model domain at each timestep on the basis of known relationships between landscape and meteorological variables. EnBal simulates the surface energy exchange according to the meteorological data distributed by MicroMet and snow evolution from the previous timestep. SnowPack evolves snow depth, density, and temperature according to precipitation input and surface conditions produced by EnBal. Last in the modelling process, SnowTran-3D redistributes snow in response to the interaction between the wind-fields at each timestep, surface topography, and vegetation snow holding capacity. SnowAssim allows the user to input in-situ or remotely sensed measurements of snow water equivalent and corrects the precipitation forcing retroactively before a second model simulation. A workflow diagram of the modelling procedure can be found in the Fig 3 in S1 Appendix.

We obtained meteorological data from the NASA Modern Era Retrospective-Analysis for Research and Applications Version 2 [MERRA-2; 71]. This gridded climate data is available hourly from 1980 to present at a resolution of 0.5˚ latitude to 0.625˚ longitude (~55 km by ~32 km). We aggregated the hourly surface forcing variables from 16 grid points covering the study domain into daily values, using the meteorological inputs required by MicroMet; temperature, relative humidity, wind speed, wind direction and precipitation. The topographic and vegetation layers required by SnowModel were derived from the Advanced Spaceborne Thermal Emission and Reflection Radiometer Global Digital Elevation Model Version 2 [ASTER GDEM; 72] and the National Land Cover Database 2011 [NLCD; 73] respectively. We conducted a simple analysis of the land cover change in each of the 9 survey units by cropping a further dataset, the NLCD 2011 Land Cover Alaska 2001 to 2011 From To Change Index [74], and analysing the extent of landcover change from 2001 to 2011. Of the 8678 $km^2$ of all 9 units, only 27 $km^2$, or 0.32%, had been classified in this dataset has having changed in landcover over the 10 year period (Table 2 in S1 Appendix). We do not believe that the rate and magnitude of this change was fast or great enough to impinge on Dall's sheep populations within the timeframe of this study, and we therefore kept land cover as a static layer in the modelling procedure. The ASTER GDEM was chosen for its complete coverage of the study domain and comparable 1-arc second resolution to the 30 m NLCD data. It was resampled (bilinear) to this resolution and reprojected into the Alaska Albers Equal Area Conic coordinate reference system to match that of NLCD. To cover the JSU, a domain of 1680 by 2244 30-m grid cells (~50 km by 67 km) was created. The 30 m resolution represents a balance between computational efficiency and the ability of the model to simulate important features of the snowscape, e.g., wind-blown areas and drifts, that occur in mountainous regions.

## Snow surveys

We obtained ground-based snow observations from September 2016 to August 2017 to calibrate and validate SnowModel. We installed 22 Reconyx Hyperfire PC900 [75] time-lapse cameras in two areas of the domain, Jaeger Mesa (~1600 m to ~2100 m elevation) and a site near Rambler Mine, Nabesna (~900 m to ~1200 m elevation). Each camera was aimed at a 1.5 m tall snow stake with bands every 5 cm, and cameras were programmed to take hourly photos (Fig 2). Camera sites were selected to capture gradients in elevation, vegetation and aspect with consideration for field safety in steep and rugged terrain. We conducted snow surveys in

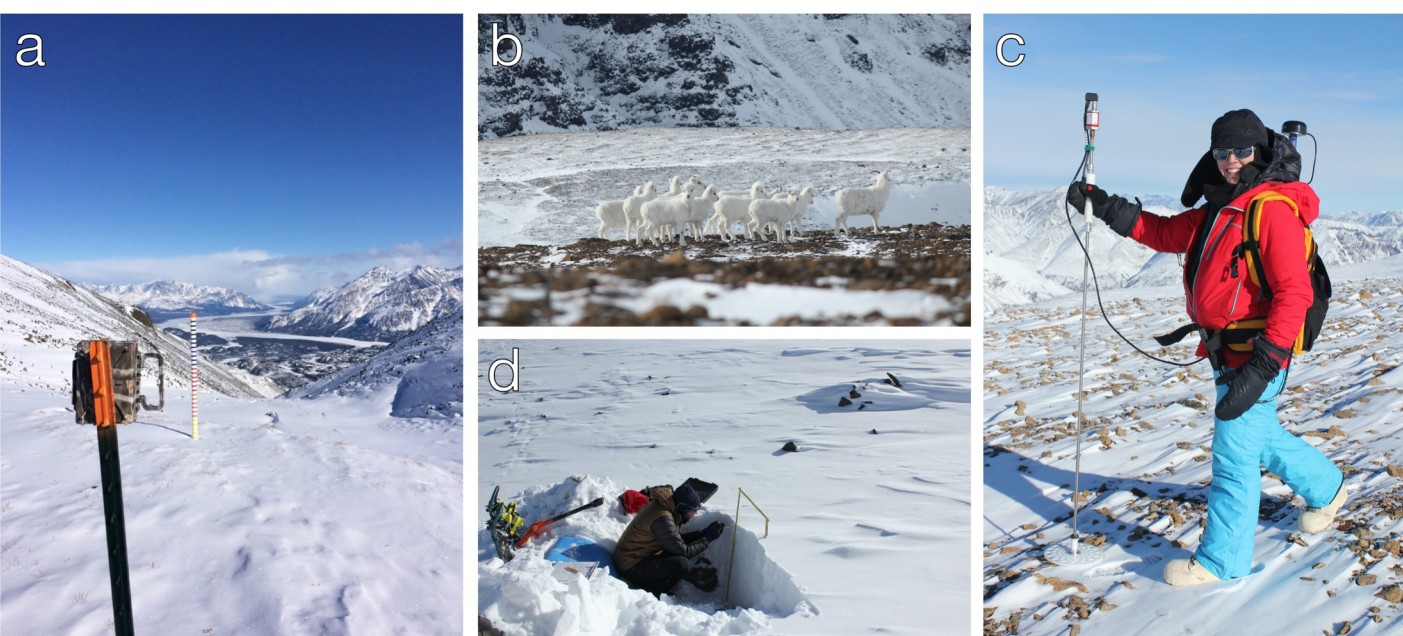

**Fig 2.** (a) Remote camera and snow stake installation looking northeast to the Nabesna river from Jaeger Mesa on 20th March 2017. Note the wind-blown, snow free areas on the slopes to the immediate sides of the snow stake. Photo C. Cosgrove. (b) Nursery band of Dall's sheep on Jaeger Mesa. (c) Laura Prugh operating the Magnaprobe to survey snow depth atop Jaeger Mesa. Photo Anne Nolin. (d) Chris Cosgrove surveying a snow pit for stratigraphy, temperature and density profile. Photo L. Prugh. The individuals in this manuscript have given written informed consent (as outlined in PLOS consent form) to publish these images.

and around the camera sites from 18[th] to 24[th] March 2017. A snow pit was excavated at a randomly selected location within 5 m of each camera and we recorded the stratigraphic profile, the temperature profile at 10 cm intervals using a digital thermometer, and the density profile double-sampled at 10 cm intervals using a Snowmetrics 1000c cutter [76] and a digital scale. For ingestion into SnowAssim, the mean density of the double sample at each interval was calculated and converted into snow water equivalent (SWE). The product of each interval's SWE was then used to calculate the bulk SWE for each pit location. In total 18 pits were possible with the remaining 4 cameras being located in areas that were snow-free. Alongside the snow pit measurements, 7806 snow depth measurements were taken and recorded using both manual and automated methods [77], with location recorded by GPS in both instances. These measurements were obtained at 2 m intervals using 4 sampling configurations: (1) 50 m transects in a cross-pattern from each camera site, (2) transects following the elevation gradient between cameras grouped by aspect on the east and west sides of Jaeger Mesa and at Rambler Mine, (3) 50 m 'spirals' randomly located on top of Jaeger Mesa, and (4) a sequence of traverses running north-to-south, east-to-west and along the edge of the northern tip of Jaeger Mesa. This sampling strategy was conducted to characterise different scales of snow-depth variability in different configurations of topography and vegetation.

## Calibration of SnowModel

A fundamental first step in improving the modelled description of snow evolution is to assess and correct the precipitation forcing ingested in the model. To do this, SnowAssim was utilised with our recorded SWE measurements in low-elevation, sparsely forested areas near Rambler Mine within a modelling run from 1[st] September 2016 to 1[st] April 2017. Using only the forested SWE data protected against error caused by assimilating SWE values from areas subject to greater wind redistribution. The synoptic scale of precipitation in the region is greater in

size than that of the modelling domain, so the precipitation accumulating in low-elevation forest areas is proxy to that falling in high elevations but is less likely to be highly redistributed by wind. A precipitation correction factor of 0.37 was found using this procedure and hence applied to the precipitation forcing from 1980 to 2017.

To reproduce the field-observed patterns of snow distribution in our model simulations, we compared snow depth, density and water equivalent field measurements from a subset of the domain to their equivalent modelled outputs. Given the focus of this study on snow conditions in Dall's sheep habitat (see below), we calibrated the model for optimum performance at high-elevations and thus used only field observations from alpine areas in this part of the calibration procedure.

Initial examination of the wind forcing data derived from MERRA-2 revealed it to be insufficiently strong to redistribute snow, a potential bias in the original data but also likely due to the suppression caused by aggregating hourly data into daily values. As snow density and wind speed interact with one another, we adjusted a scalar increasing the windspeed in the meteorological forcing data and a SnowModel parameter controlling the snow density evolution together. After an initial sensitivity analysis, our calibration involved 72 SnowModel simulations from 1$^{st}$ September 2016 to 1$^{st}$ April 2017 with the density adjustment factor ranging from 2.0 to 10.0 in increments of 1.0, and the wind speed scalar ranging from 1.5 to 5.0 in increments of 0.5. To establish the best calibration, each snowpit-observed bulk snow density measurement was compared to the modelled bulk snow density in the equivalent model grid-cell and the Root Mean Squared Error (RMSE) was computed. Using the same procedure, observed snow depth was compared to modelled snow depth, with observed snow depths being aggregated into a mean value for each grid cell given the high resolution of our depth surveys. Additionally, for the grid cells where bulk density was available, we compared observed SWE to modelled SWE. RMSE values for density, depth, and SWE were ranked among the 72 simulations, and the mean ranking of each simulation was then calculated. The parameters from the top-ranked calibration were then used to model snow properties for the study domain from September 1$^{st}$ 1980 to August 31$^{st}$ 2017. To further test the calibration, a validation was conducted using the snow depths acquired from the remote camera installations.

## Model derived covariates

To limit our modelled snow properties to Dall's sheep habitat, we selected only pixels that correspond to their preferred land cover above 1200 m. Roffler et al. (see see supplementary materials RSF_S3.png in [78]) found this elevation to be the lower limit of Dall's sheep core habitat in WRST using locations of sheep observed during surveys, albeit for summer months. To delineate the land cover that Dall's sheep select for, we included only pixels corresponding to the Dwarf Shrub and Barren Land classifications in the NLCD product [73]. This follows numerous studies that have found that Dall's sheep select for open, sparsely vegetated areas at mid- to high-elevations [e.g., 55], and recent habitat selection models driven by GPS-collar data have confirmed this [65]. We recognise that Dall's sheep may use lower elevations in winter, but there are no currently published data describing their winter distribution in our study region.

Four snow covariates were derived for comparison to the following summer's lamb-to-ewe ratios: mean snow depth, mean snow density, total snowfall and percent 'forageable area'. Additionally, we included SnowModel-derived mean air temperature as a climate covariate. For mean snow depth, mean snow density, total snowfall, and mean air temperature, the daily mean was found for all grid cells matching the above criteria first. Seasonal means

(fall = September, October and November; winter = December, January and February; spring = March, April and May) were then calculated from the daily data in the case of mean snow depth, mean snow density and mean air temperature, whereas the daily data was summed by season for total snowfall. Higher incidences of snow depth, snow density and snowfall were expected to be deleterious to Dall's sheep productivity, with increases in air temperature anticipated to lead to increases in lamb-to-ewe ratios. The final covariate, mean percent 'forageable area', was calculated as the seasonal mean of the daily percentage of Dall's sheep habitat with snow depth beneath half-chest height (0.25 m) and snow density beneath 330 kg m$^{-3}$. These snow conditions were found by Mahoney et al. [65], and confirmed in the field by Sivy et al. [79], to be selected by Dall's sheep at movement scales typical of foraging behaviour. We hence expected greater percentages of forageable area to correlate with increased Dall' sheep productivity. To test whether there was a delayed effect from conditions in the previous snow season to parturition, i.e. >1 yr previous to the summer of sheep survey, we also calculated aggregate metrics for all of the above variables for both the previous summer (reported as 'Previous Summer') and all months where snow cover is a dominant feature in the study area (September through May, reported as 'Previous Year').

## Statistical analyses

To examine the relationships between the model derived snow and climate metrics and lamb-to-ewe ratios, we employed multiple regression models after a covariate selection process. All analyses were conducted in the R program [80]. As a first step we tested whether the inclusion of Survey Unit as a random effect was significant in models using each of our seasonal snow and climate covariates as a single predictor. To do this we used ANOVA to test for significant difference between paired models of the same predictor but fitted with and without Survey Unit as a random effect using the R package nlme [81]. At this step, all models were fitted using restricted maximum likelihood (REML) to allow for valid comparison between the model with and the model without the random effect [82] and we additionally tested a null model. We then ranked each single predictor model and the null model, when fitted without the addition of the random effect term and using Ordinary Least Squares (OLS), by their second-order Akaike Information Criterion (AICc). Covariates that were found to be ranked higher than the null model as single predictors were subsequently considered as additional additive terms in multiple regression models. To avoid over-parameterization on a small dataset we restricted the number of predictors per model to three and excluded any covariates that had a collinearity of greater than 0.7 in the same model. The final list of single- and multi-predictor models was finally ranked by their AICc to discern which snow and climate covariates had the greatest explanatory power in isolation or combination. Linear regression was used to test for trends in covariate values from 1980 to 2017 by season. Likewise, the coefficient of variation (CV) was calculated for a rolling 10-year window for each snow and climate metric, and linear regression was used to test whether the degree of interannual variability increased over time. An alpha of 0.05 was used for evaluating statistical significance throughout, with the exception of testing each productivity model's intercept and predictor estimates, where a Bonferroni-corrected alpha level, as calculated by alpha divided by the total number of models in the final list, is reported to reduce the chance of type 1 error.

## Results

### SnowModel calibration

The parameter combination that best produced our observations of depth, density and SWE was a density adjustment factor of 6.0 and a wind speed increase of 2.5, producing RMSEs of

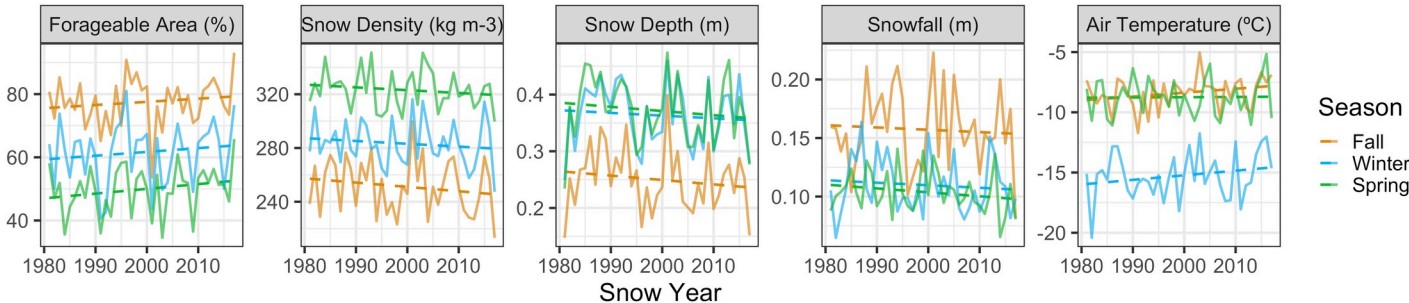

**Fig 3. Time series and trends of each snow and climate covariate by season 1980 to 2017 within the Jacksina sheep survey unit within Wrangell-St. Elias National Park and Preserve, Alaska.** Note the similar pattern of snow depth year-on-year across all three seasons and the close similarity of the mean snow depth in winter and spring.

0.09 m snow depth, 31.71 kg m⁻³ snow density and 0.04 m SWE (Fig 4 and Table 3 in S1 Appendix). Taking the snow depth from images recorded daily at 12:00 Alaska Standard Time, there were 4996 available days of data from 17 cameras located outside of forested and shrub areas. Comparison of the camera snow depth to model snow depth yielded an RMSE of 0.08 m, which is comparable to that from the spatial calibration albeit with an average 0.06 m bias towards over estimation (Fig 5 in S1 Appendix).

## Summary of sheep surveys and modelled snow and climate metrics

Of the 41 surveys across 19 years included in the analysis (Table 1 in S1 Appendix), the mean lamb-to-ewe ratio was 0.30 (±0.10 SD), with a maximum of 0.55 sampled in the Mount Drum survey unit in 1981 and a minimum of 0.09 sampled in Jacksina in 1993. Snow depths in fall (mean = 0.28 m ±0.06 SD) were always lower than both winter (mean = 0.40 m ±0.06 SD) and spring (mean = 0.42 m ±0.07 SD), which generally had a similar mean snow depth and closely followed the interannual variability established in fall (Table 4 in S1 Appendix; Fig 3).

## Model derived covariates and lamb-to-ewe ratios

The addition of Survey Unit as a random effect was not shown to be significant for any of the initial single predictor models (see Table 5 in S1 Appendix). Hence, we continued our model selection with models fitted by OLS. When comparing the single predictor models of each snow and climate covariate 11 models were ranked higher by AICc than the null model (see Table 5 in S1 Appendix), none of which contained a covariate pertaining to the previous year's Summer or snow season indicating that there wasn't a delayed-effect from the previous snow season. Of the covariates ranked higher than the model only snowfall (fall, winter, and spring in order of weighting) and air temperature (fall) were found to be under the cut-off for collinearity. Fall snowfall and fall air temperature were therefore used in two and three predictor linear models in combination with the other covariates leaving 40 models, inclusive of the null model, in our final list (see Table 6 in S1 Appendix).

Of the top ranked models, 5 are shown to be well supported (ΔAICc < 2) and all include a seasonal covariate of snow depth and fall air temperature in their predictors (Table 1). The highest ranked model, fall snow depth and fall air temperature has an adjusted R-squared of 0.41 and is significant to the Bonferroni-corrected alpha level for the intercept and fall snow depth terms, and alpha for fall temperature (Table 1). Coefficients from this model indicate that increases in fall snow depth and decreases in fall air temperature lead to a decline in the following summer's lamb-to-ewe ratio (Fig 4). All models that contain snow depth as a term

**Table 1. Top 10 models as ranked by second order Akaike Information Criterion (AICc).**

| Model | Intercept (SE) | 1st Predictor Estimate (SE) | Fall Air Temperature (SE) | Fall Snowfall (SE) | K | Delta AICc | AICc weight | R-Sq. | Adjusted R-Sq. |
|---|---|---|---|---|---|---|---|---|---|
| Fall Snow Depth + Fall Air Temperature | 0.690 (0.094)** | -0.738 (0.193)** | 0.027 (0.012)* | – | 3 | 0 | 0.188 | 0.439 | 0.41 |
| Winter Snow Depth + Fall Air Temperature + Fall Snowfall | 0.900 (0.112)** | -0.599 (0.214)* | 0.032 (0.012)* | -0.818 (0.398)* | 4 | 0.134 | 0.176 | 0.472 | 0.429 |
| Spring Snow Depth + Fall Air Temperature + Fall Snowfall | 0.851 (0.111)** | -0.522 (0.192)* | 0.027 (0.013)* | -0.940 (0.388)* | 4 | 0.523 | 0.145 | 0.467 | 0.424 |
| Fall Snow Depth + Fall Air Temperature + Fall Snowfall | 0.780 (0.114)** | -0.593 (0.219)* | 0.030 (0.012)* | -0.593 (0.435) | 4 | 0.592 | 0.14 | 0.466 | 0.423 |
| Winter Snow Depth + Fall Air Temperature | 0.792 (0.103)** | -0.738 (0.211)** | 0.029 (0.012)* | – | 3 | 1.963 | 0.071 | 0.412 | 0.381 |
| Fall Snow Depth | 0.511 (0.046)** | -0.895 (0.187)** | – | – | 2 | 2.328 | 0.059 | 0.37 | 0.354 |
| Spring Snow Depth + Fall Snowfall | 0.689 (0.081)** | -0.720 (0.173)** | – | -0.848 (0.402)* | 3 | 2.374 | 0.057 | 0.406 | 0.375 |
| Spring Snow Depth + Fall Air Temperature | 0.706 (0.099)** | -0.623 (0.199)* | 0.023 (0.014) | – | 3 | 3.949 | 0.026 | 0.383 | 0.35 |
| Fall Snow Depth + Fall Snowfall | 0.552 (0.068)** | -0.818 (0.211)** | – | -0.367 (0.452) | 3 | 4.084 | 0.024 | 0.381 | 0.348 |
| Spring Snow Depth | 0.577 (0.064)** | -0.788 (0.177)** | – | – | 2 | 4.45 | 0.02 | 0.336 | 0.319 |

Standard error (SE) shown in brackets for both the intercept and estimate of each predictor in each model. 1st Predictor indicates the 1st snow and climate covariable listed in the Model column.

** indicates significance at a Bonferroni corrected alpha level of 0.00125 (alpha / total models)

* indicates significance at alpha = 0.05. P-values were computed in R by the Wald test method via use of the 'summary' core package [80].

outperform models using other snow and climate metrics (see Table 6 in S1 Appendix). Estimates of snow depth, snow density and snowfall in all models indicate that increases in these variables decreased lamb-to-ewe ratios, whereas estimates for air temperature and forageable area showed a positive relationship between these predictors and lamb-to-ewe ratios, following expected relationships (Table 1, Table 6 in S1 Appendix).

## Trends and variance in seasonal covariates 1980 to 2017

No statistically significant trends were found for modelled snow metrics from 1980 to 2017 (Table 7 in S1 Appendix; Fig 3). However, model estimates indicated decreasing snowfall, snow depth and snow density, and increasing forageable area and air temperature for all seasons (Fig 3). The interannual variation in winter snow density significantly increased during the time series (Table 8 in S1 Appendix; Fig 5). In contrast, winter snowfall was found to be significantly less variable over time (Table 8 in S1 Appendix; Fig 5). The highest interannual CVs (non-rolling) occurred in fall for both snow depth (CV = 22.21%) and snow density (CV = 8.06%), winter for both snowfall (CV = 21.87%) and forageable area (CV = 15.46%), and spring for air temperature (CV = 17.36%; Table 4 in S1 Appendix).

## Discussion

The impact of changing snow conditions on wildlife in northern ecosystems is of both ecological and societal concern as these remote regions are signalling major impacts of accelerated warming [83]. However, studies are limited by data that are scarcely distributed in time and space in the region, especially in alpine areas [27], and there remains uncertainty as to when

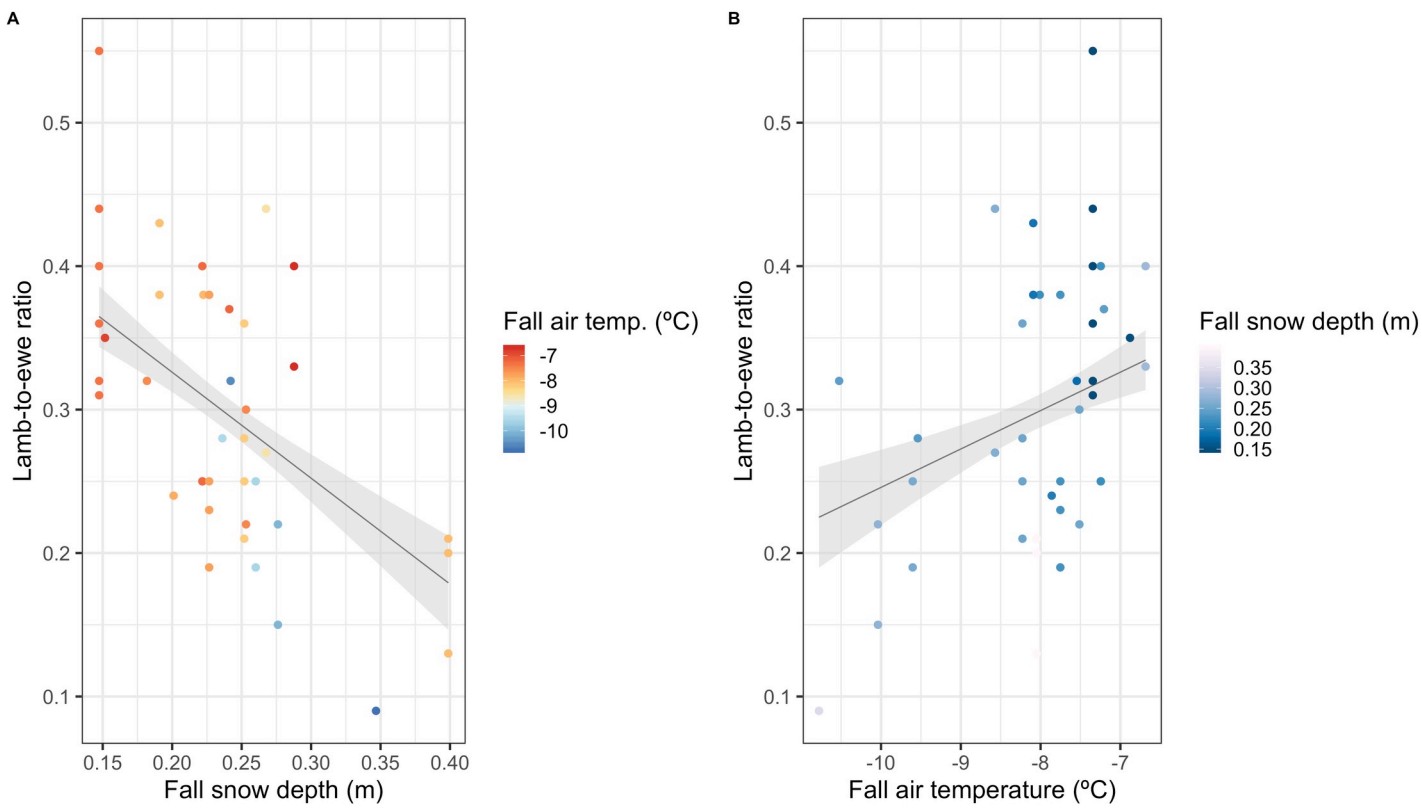

**Fig 4.** A) An increase in fall mean snow depth decreases Dall's sheep summer productivity, here defined as lamb-to-ewe ratio, whereas B) increased fall mean air temperature increases summer productivity. Estimates and the shaded grey 95% confidence interval are derived from the top model as ranked by AICc in Table 3.

and what snow conditions are most important to wildlife demography. Here we use a spatially distributed snow model to simulate snow and climate conditions over 37 years in the northern Wrangell-St Elias National Park and Preserve (WRST) to better understand the influence of snow properties on the dynamics of Dall's sheep. Snow conditions, most notably increased snow depth, were strongly associated with declines in Dall's sheep productivity, with decreased air temperature and increased snowfall also leading to decreased lambs being observed in summer, though with less predictive power in comparison to snow depth. Our top-ranked model (s) indicated that fall was the time period that these snow and climate conditions were most

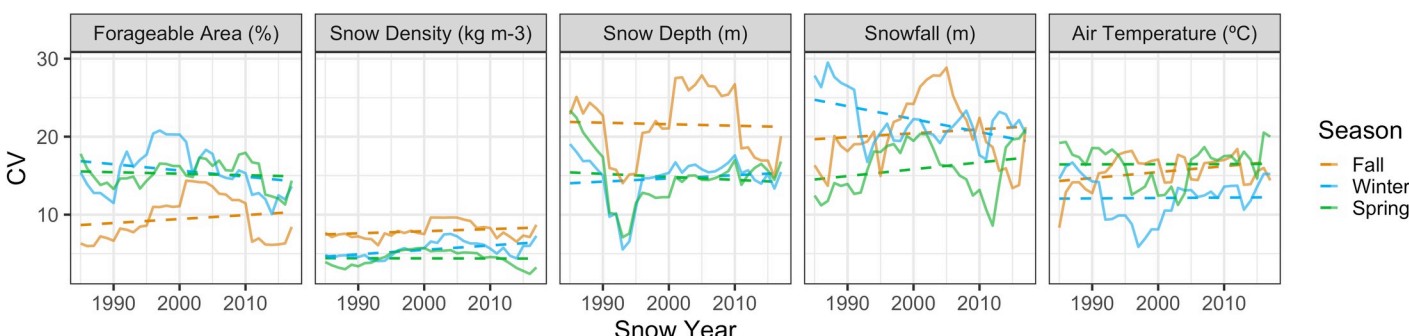

**Fig 5. Time series of 10-year rolling coefficient of variability (CV) for each snow and climate covariate by season within the Jacksina sheep survey unit within Wrangell-St. Elias National Park and Preserve, Alaska.**

important. These findings suggest that challenging snow conditions that persist throughout the snow year, as per our first hypothesis, are more important to Dall's sheep productivity than conditions during the spring lambing season, as described by our second hypothesis.

Similar to other alpine and Arctic ungulates, Dall's sheep access forage by either 'cratering', wherein they dig through the snow, or by finding snow-free areas. Deeper snow has been shown to reduce foraging efficiency in studies of other ungulates, potentially leading to increased caloric deficit and decreased birth mass in offspring [48, 84]. Thus, early establishment of deep snow conditions may lead to energetically challenging conditions over many months, protractedly decreasing the body condition of ewes and therefore decreasing their ability to successfully produce, protect and nurse healthy lambs in the weeks immediately after birth. The significance of Fall air temperature as an additional term in the top models further suggests that early-season calorific expenditure, through the increased cost of thermoregulation in this instance [85], is more damaging to productivity than that occurring closer to lambing.

Two recent large-scale studies stand in contrast to our results. Van de Kerk et al. [47] and Rattenbury et al. [42] found that the date of snow disappearance best predicted Dall's sheep productivity, with later dates resulting in lower lamb-to-ewe ratios. Both studies noted that this relationship was weaker at lower latitudes, including that of WRST, and suggested the comparatively extended growing season in these ranges may buffer the effect of severe winters due to increased forage abundance and quality. However, van de Kerk et al. [47] additionally found that snow cover duration, i.e. the number of days snow is present each winter, also had an effect on lamb-to-ewe ratios, albeit relatively weaker than snow disappearance date, and hence proposed that extended exposure to difficult conditions is less important than the snow cover immediately before or after to lambing. Snow disappearance dates depend on the energy balance of a snowpack, along with weather conditions and other variables. Thin, low density snow cover can extend later into the year if air temperatures are cool enough to preserve it, while deep, dense snows can rapidly disappear due to early spring conditions with high temperatures and rain [86]. Hence, inference of the vertical properties of snow from its extended presence in remote sensing data is not always reliable and cannot describe the evolution of snow depth and density throughout the entirety of a snow season. Our methods here highlight the importance of vertical snow properties on northern wildlife such as Dall's sheep and show that detailed, local analyses of snow properties can reveal new insights that range-wide remote sensing methodologies, such as van de Kerk et al. [47] and Rattenbury et al. [42], may not be able to detect. Our results also compare well statistically; while van de Kerk et al. [47] do not report comparable metrics, for the Nabesna area within their analysis, which is within our study area, Rattenbury et al. [42] report an R-squared of 0.33 [42; see Fig 4], which is lower than our top model's adjusted R-squared of 0.41.

The effects of snow on the movement, habitat selection, and energetics of various wildlife has been relatively well studied [27], but there is a lack of evidence linking the impact of snow conditions on fine-scale behavior to broad-scale demographic consequences [65]. Mahoney et al. [65] found that Dall's sheep in Lake Clark National Park strongly favoured areas of less dense, shallow snow at fine-scale movements associated with foraging, illustrating that habitat selection is affected by snow density as well as depth. Forageable area, a variable derived from the area available below a threshold density and snow depth found in Mahoney et al. [65], showed relatively poor predictive power (Table 5 in S1 Appendix). This was unexpected given the forageable area metric's increased detail and foundation in field observations [79]. However, we suggest that an explanation for this might be that the *actual* forageable area is quite different from the *modelled* forageable area. For example, low-snow or snow-free areas might be devoid of forage or, even if forage is present, these areas might be in terrain that is avoided

by Dall's sheep due to predation risk. Mean snow depth, conversely, is highly ranked for all seasons and is possibly a more reliable metric for describing the relative efficiency of winter foraging behaviour.

Here we have focused on the impact of snow conditions on Dall's sheep productivity. However, it is important to note that productivity and survival are influenced by additional factors, including predation and interspecific population dynamics [43, 45, 87], forage quantity and quality [44, 45], and in rare cases by disease [87]. Other mountain ungulates have shown declines in productivity in response to high population densities and climactic forcing [e.g., 88–91]. However, a preliminary study of a simple regression of density (as calculated by the total number of surveyed adult sheep, inclusive of yearlings, divided by the area of the Survey Unit) vs lamb-to-ewe ratios in our dataset did not show any relationship suggesting density dependence was not important in our study area. This follows the findings of van de Kerk et al. [47; see Appendix 2] that found no effect of the survey date and population density on lamb-to-ewe ratios and used data from a much larger, range-wide dataset of 534 surveys. However, habitat-selection models of Dall's sheep, e.g. Roffler et al. [78], suggest that Dall's sheep likely utilize only certain locations of the Survey Units they are reported within, e.g. areas predominantly near escape terrain and devoid of tall vegetation. Hence, the simple calculation of density described above, and used by van de Kerk et al. [47], is likely to be prone to underestimation and vary in accuracy according to the relative abundance of preferred habitat in each Survey Unit. We therefore suggest that further work incorporates insights from habitat selection modelling to better test for any density dependence on productivity in Dall's sheep.

In response to other studies that show a lagged effect of snow and climate conditions on the body condition and parturition rate of other ungulates [e.g., 91, 92] we tested the importance of the previous summer's and the previous snow season's snow and climate conditions on productivity. No significant relationships were found (Table 5 in S1 Appendix), suggesting that the snow and climate conditions for the season immediately before lambing are more important for productivity. Our dataset however does not include variables pertaining to the quality of vegetation available to ewes in the summer preceding or current to lambing. Both early [93] and more recent work [94] has connected metrics of summer forage quality with both lamb survival rates [93, 94] and Dall's sheep productivity [93]. Also beyond the scope of the current study are the effects of interspecific relationships. The primary predators of Dall's sheep, coyote (*Canis latrans*) and golden eagles (*Aquila chrysaetos*), have been shown to account for less lamb mortality in summers with a high Normalized Difference Vegetation Index (NDVI) [94] and are likely to prey more on Dall's sheep during years with low snowshoe hare numbers [43, 45]. To gain a more holistic understanding of Dall's sheep productivity and population dynamics, attention needs to be paid to a wide range of biotic and abiotic factors that are not considered here. The adjusted R-squared of our top ranked model with only snow properties included (fall snow depth, R-sq. = 0.35; Table 1), is likely indicative of our narrow focus. However, our findings do illustrate that snow properties, and in particular their early establishment, are important factors for Dall's sheep productivity and stand to inform further research into population dynamics of Dall's sheep and other wild ungulates.

Seasonal snow throughout the northern hemisphere is being altered in terms of its coverage, timing, duration and physical properties as a response to climate change [7]. The increase in extreme events, such as late snow disappearance in spring 2013 in Alaska, are considered a likely product of climate change that might impinge on Dall's sheep productivity [95]. However, we found no evidence that snow conditions important to Dall's sheep productivity have markedly changed in WRST from the long-term mean or have increased in terms of interannual variability during our study period. This may be due to the sub-Arctic location of

northern WRST in Alaska's dry interior where changes to the form and volume of precipitation are less pronounced than in wetter and warmer maritime regions [7].

Verbyla et al. [60] noted substantial differences in climate and snowline elevation throughout Dall's sheep ranges and found that the mean snow line elevation on May 15th had pronounced interannual variability in the central and eastern Brooks Range. It is in these Arctic Alaskan ranges that are on the fringe of suitable Dall sheep habitat where the greatest population decreases in Dall's sheep have been observed, prompting emergency harvest closures in some areas [36]. Dall's sheep sensitivity to spring snow conditions at these high latitudes has been established by van de Kerk et al. [47] and Rattenbury et al. [42], and it may be that higher interannual variability in the elevation of spring snow line, potentially indicating a greater frequency of extreme events, is responsible for the recent declines in Dall's sheep populations in these areas [42, 47, 60]. Dall's sheep populations in *sub*-Arctic ranges in Alaska, including WRST, have population trends that are generally regarded as being stable, with the exception of the maritime Kenai peninsula [36]. If the impact of climate change on snow conditions in these ranges has yet to be acute, such as in the case of our results, it is possible that low-latitude interior mountain ranges may represent refugia for Dall's sheep and other snow-influenced alpine species [96]. Wildlife populations, particularly those that have low reproductive rates like Dall's sheep, may be resilient to sporadic extreme conditions but become vulnerable if extreme conditions become more frequent [97]. Hence, further work examining regional, long-term trends in the interannual variability of snow conditions would prove valuable in determining where climate change poses the greatest threat to alpine wildlife populations.

Our modelling approach combined with several decades of survey data demonstrated seasonal variation in the impact of snow conditions on Dall's sheep productivity in Wrangell-St Elias National Park and Preserve. However, some caution should be exercised when extending our results to other regions given the specificity and assimilation of in-situ data from our study area. While our methodological approach yields novel insights regarding seasonal snow properties in comparison to alternative approaches using optical remote sensing datasets, it also comes with its own inherent disadvantages, including limited spatial coverage, high computational demand, necessity of technical expertise, and inherent uncertainties when modelling a physical environment. Although we conducted intensive field surveys to improve the calibration of our model, these surveys occupied a small spatial and temporal extent within the larger modelling domain. This is despite efforts made to sample a wide representation of elevation, aspect and landcover during snow surveys and the installation of remote cameras. With data lacking to test the model against in-situ measurements from previous years it is possible that the model is only representative to its calibration year. While this is an important source of uncertainty, the small RMSE and bias shown in our calibration and temporal validation results does suggest our approach has promise in long-term studies of other wildlife, especially so where there are in-situ, long-term snow and meteorological datasets for model-forcing and assimilation.

## Conclusions

The establishment of a deep snowpack in fall alongside low fall temperatures was found to best explain decreased Dall's sheep productivity during the following summer. An incremental effect of season-long environmental conditions on ewe body condition hence appears to be of greater importance than spring snow conditions in our study area, a finding contrary to studies based on snow cover rather than depth [42, 47]. Our results potentially demonstrate an important link between known fine-scale effects of snow conditions, i.e. selection of shallow and/or less dense snow, with broad-scale patterns of demography. We hence propose that our

utilization of a spatially distributed snow model has scope for application in studies of other snow-influenced wildlife. Though additional data that establishes direct links between snow properties, animal movements and body condition, forage opportunity, and infant survival rates are needed for a complete mechanistic understanding of snow impacts. We found no significant trends in the long-term mean, or in a rolling measure of interannual variation, of modelled snow properties that were shown to be important to Dall's sheep productivity. Climate change hence appears to not yet be having a strong effect on snow conditions in our study domain, a result that is of broader ecological interest. However, if climate change does lead to major changes in future snow depths, our findings indicate that Dall's sheep productivity may be strongly affected.

## Supporting information

**S1 Appendix. Document containing supporting figures and tables listed in manuscript.**
(PDF)

**S1 Data. Table containing raw data of northern Wrangell St Elias Sheep surveys.**
(CSV)

**S2 Data. Table containing model derived snow covariates for each snow year from 1981 to 2017.** Where; snod = snow depth, sden = snow density, forage = forageable area, and spre = snowfall, tair = air temperature.
(CSV)

**S3 Data. Table combing sheep surveys and snow covariates.**
(CSV)

**S4 Data. Snow depth derived from SnowModel.**
(CSV)

**S5 Data. Snow density derived from SnowModel.**
(CSV)

**S6 Data. Snowfall derived from SnowModel.**
(CSV)

**S7 Data. Forageable area derived from SnowModel.**
(CSV)

**S8 Data. Air temperature derived from SnowModel.**
(CSV)

**S1 File. Python script combining snow variable data with sheep survey data.**
(PY)

**S2 File. Python script compiling snow covariate data by season from 1981 to 2017.**
(PY)

**S3 File. Python script compiling snow and sheep data together.**
(PY)

**S4 File. R script compiling Fig 3 and Table 7 in S1 Appendix.**
(R)

**S5 File. R script compiling Fig 5 and Table 8 in S1 Appendix.**
(R)

**S6 File. Zip file containing data, scripts and instructions to create Fig 2 in S1 Appendix.**
(ZIP)

**S7 File. Zip file containing data, scripts and instructions to create Fig 4 in S1 Appendix.**
(ZIP)

**S8 File. Zip file containing data, scripts and instructions to create Fig 3 in S1 Appendix.**
(ZIP)

**S9 File. R script compiling Tables 5 and 6 in S1 Appendix and Table 1 and Fig 4 in manuscript.**
(R)

**S10 File. Zip file containing data, scripts and instructions to create Fig 3 in S1 Appendix.**
(ZIP)

## Acknowledgments

Major thanks go to Glen Liston for assisting in the calibration and running of SnowModel. The Ellis family in Nabesna, AK and 40 Mile Air in Tok, AK greatly aided field logistics. Kelly Sivy and Anika Pinzner assisted with the field surveys. We thank the ADF&G, NPS and contracted (e.g. pilots) personnel who conducted the sheep surveys. Comments and insight from the academic editor and 3 reviewers appreciably improved the paper.

## Author Contributions

**Conceptualization:** Christopher L. Cosgrove, Anne W. Nolin, Laura R. Prugh.

**Data curation:** Christopher L. Cosgrove, Jeff Wells, Judy Putera, Laura R. Prugh.

**Formal analysis:** Christopher L. Cosgrove.

**Funding acquisition:** Anne W. Nolin, Laura R. Prugh.

**Investigation:** Christopher L. Cosgrove.

**Methodology:** Christopher L. Cosgrove, Jeff Wells, Anne W. Nolin, Judy Putera, Laura R. Prugh.

**Project administration:** Christopher L. Cosgrove, Anne W. Nolin, Laura R. Prugh.

**Resources:** Jeff Wells, Anne W. Nolin, Laura R. Prugh.

**Software:** Christopher L. Cosgrove.

**Supervision:** Anne W. Nolin, Laura R. Prugh.

**Validation:** Christopher L. Cosgrove.

**Visualization:** Christopher L. Cosgrove, Laura R. Prugh.

**Writing – original draft:** Christopher L. Cosgrove.

**Writing – review & editing:** Christopher L. Cosgrove, Jeff Wells, Judy Putera, Laura R. Prugh.

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
