## [Decision Letter · Decision Letter 0]

9 Apr 2020

PONE-D-20-03620

Seasonal influence of snow conditions on Dall’s sheep productivity in Wrangell-St Elias National Park and Preserve

PLOS ONE

Dear Mr Cosgrove,

Thank you for submitting your manuscript to PLOS ONE. After careful consideration, we feel that it has merit but does not fully meet PLOS ONE’s publication criteria as it currently stands. Therefore, we invite you to submit a revised version of the manuscript that addresses the points raised during the review process. Reviewer #2, for example, pointed out that your manuscript is well structured and written but it could be slightly shortened by cutting certain sentences and avoiding excessive repetition of the same information in different sections. The reviewer #1, however, recommend the inclusion of the interactive effect of snow metrics with a proxy of population density (which should include yearling, rams and ewes, but not lambs), possibly with a delayed effect. In fact, it is well known that snow might affect demography of mountain ungulates with a delayed effect. In my own experience, we observed strong delayed effects of snowfalls on body stores, i.e., in years with a lot of snow Ibexes body stores reached their lowest values in the current winter–spring, but the highest in the following summers and autumns (Serrano et al. 2011. Eur J Wildl Res. 57:45–55). That example could also be applied to reproductive parameters such as Dall's sheep productivity. The third reviewer underlined the need of a reference to the environmental scenarios in which the SnowModel has been tested and validated. This reviewer also recommend providing more consideration of its appropriate application to different latitude settings.

We would appreciate receiving your revised manuscript by May 24 2020 11:59PM. To enhance the reproducibility of your results, we recommend that if applicable you deposit your laboratory protocols in protocols.io, where a protocol can be assigned its own identifier (DOI) such that it can be cited independently in the future. For instructions see: http://journals.plos.org/plosone/s/submission-guidelines#loc-laboratory-protocols

We look forward to receiving your revised manuscript.

Kind regards,

Emmanuel Serrano, PhD

Academic Editor

PLOS ONE

2. In your Methods section, please provide additional location information of the study area, including geographic coordinates for the data set if available.

4. We note that Figure 1 and S1 in your submission contains map images which may be copyrighted. All PLOS content is published under the Creative Commons Attribution License (CC BY 4.0), which means that the manuscript, images, and Supporting Information files will be freely available online, and any third party is permitted to access, download, copy, distribute, and use these materials in any way, even commercially, with proper attribution. For these reasons, we cannot publish previously copyrighted maps or satellite images created using proprietary data, such as Google software (Google Maps, Street View, and Earth). For more information, see our copyright guidelines: http://journals.plos.org/plosone/s/licenses-and-copyright.

1.    You may seek permission from the original copyright holder of Figure 1 and S1 to publish the content specifically under the CC BY 4.0 license. 

5. We note that Figure 2 includes an image of a participant in the study.

Reviewers' comments:

Reviewer's Responses to Questions

**Comments to the Author**

1. Is the manuscript technically sound, and do the data support the conclusions?

Reviewer #1: Partly

Reviewer #2: Yes

Reviewer #3: Yes

2. Has the statistical analysis been performed appropriately and rigorously? 

Reviewer #1: I Don't Know

Reviewer #2: Yes

Reviewer #3: Yes

3. Have the authors made all data underlying the findings in their manuscript fully available?

Reviewer #1: Yes

Reviewer #2: Yes

Reviewer #3: Yes

4. Is the manuscript presented in an intelligible fashion and written in standard English?

Reviewer #1: Yes

Reviewer #2: Yes

Reviewer #3: Yes

5. Review Comments to the Author

Reviewer #1: General comments

My critical review of Cosgrove et al.’s paper on Seasonal influence of snow conditions on Dall’s sheep productivity will be fairly short, as ~70% of the paper (at last in the methods section) deals with subjects (i.e. remote sensing, climatic analyses) that are out of my area of expertise (wildlife/ungulate biology). I would thus recommend the editor to base his evaluation also on the comments of experts in these other fields.

That said, the MS appears generally well written, with mostly appropriate analyses (but see comments below) and suitable for the readership of PLoS ONE. After some adjustments, I think it will represent a nice addition to the existing literature on mountain ungulate ecology.

Specific comments

l. 116-128: before these paragraphs you have correctly discussed several snow characteristics that can have an impact on sheep (and wildlife in general), but then here you use the somewhat fuzzy term ‘snow conditions’. What does that mean? Hardness? Persistence? Depth?

l. 190-195: so, the sheep counts were not spatially explicit, or? I am not sure I understand at which spatial scale you coupled counts and snow cover conditions.

l. 195-204: I am not sure I understand here. So, lambs have been clearly identified I suppose, and ‘ewe-like’ counts have been pooled together with ‘true’ ewe counts, even though they might have been yearlings (of either sex) and/or young rams, right? If I misunderstood, please clarify. If I understood correctly, I am wondering why you would need to include them in the denominator in the first place. Can’t you simply use the ‘true’ ewe counts as a more robust (less error-prone) proxy of lamb-to-ewe ratio? Non-mature individuals and rams do not contribute to lambing anyway, so why would you include them in the counts?

l.349-369: The stats are all in all alright, but can be much improved. First, it is not clear to me why you would use total counts as a weight in your model. Did you have issue of heteroskedasticity? Second, and most importantly, using only snow metrics as predictors is rather simplistic. Mountain ungulate demography is well known to be affected by the synergistic effect of climate and density (see Jacobson et al. 2004 or Corlatti et al. 2019). So, at least I would have expected the inclusion of the interactive effect of snow metrics with a proxy of population density (which should include yearling, rams and ewes, but not lambs), possibly with a delayed effect. Just looking at your data in Table 1 I suspect (but I might be wrong) that some density-dependent effect may be present. Other climatic features might also play a role, temperature for instance. The main point here is that snow may surely be the single most important parameter driving sheep dynamics, yet you ideally need to provide some more support for that (i.e. you need to convince the reader that no other parameter is driving the lamb-to-ewe ratio). Third, I am not sure why you did not test for at least a 1-year delayed effect of snow: the lamb-ewe ratio may well depend on the fact that the mother was not fertilized in the first place, because of poor body conditions, which might depend on the snow conditions of the year before.

l. 396-407: a test and a measure of goodness of fit of your selected linear model are missing. Did you inspect model residuals? Can you provide a measure of R2 (marginal and conditional)?

In absence of clarifications with respect to potentially different drivers (i.e. density dependence, delayed effects on lamb-to-ewe ratio) is difficult to evaluate the Discussion.

References

Jacobson, A.R., Provenzale, A., von Hardenberg, A., Bassano, B. & Festa-Bianchet, M. (2004). Climate forcing and density dependence in a mountain ungulate population. Ecology 85, 1598–1610.

Corlatti, L., Bonardi, A., Bragalanti, N. and L. Pedrotti. 2019. Long-term dynamics of Alpine ungulates suggest interspecific competition. Journal of Zoology 309, 241–249.

Reviewer #2: To question 1: I answered yes, but there are, a few instances where there was over-interpretation of the results (e.g. AICc model selection).

To question 2: I answered yes, but the concerns mentioned for question 1 are still valid. I think the authors have mostly used the statistical models presented in the manuscript in an appropriate fashion. However, they should have used a different approach to confronting multi-collinearity. Also, I am not an expert on the SnowModel and thus cannot judge this section.

To question 3: I answered yes because the data is available.

To question 4: I answered yes because the manuscript is well-written and in standard English. However, I think the manuscript could be shortened with more concise writing.

Additional comments are in an attached file.

Reviewer #3: This is a very relevant modelling study conducted to link ungulate productivity with environmental constraints on the landscape, particularly in relation to snow cover and other associated variables. Use is made of remote sensing data to establish snow cover extent and timings and to verify topographical distribution of key study areas. Use is made of a long time series of Dall Sheep population data to establish relationships between these population numbers, particularly ewe to lamb ratios, and modelled snow cover over the duration of the study using a year of environmental data to constrain the model. The suitability of the model is determined through RMSE analysis relating to the in situ data with snow depth from remote cameras used as the independent verification source.

Abstract

The study is well outlined in the abstract. There is clear emphasis of the research question to be addressed as well as providing some descriptions of the results. Quantitative reporting of the regression and statistical key results would be welcome here in more detpth. Another thing to consider in the abstract is to report the RMSE of the model to the snow characteristics you are trying to replicate. This is an essential part of the study. Additionally it would be useful to discount the role of temperature in these relationships at this stage so that the focus is clearly established on snow cover independent of temperature.

Introduction

The hypotheses of the study are well presented at the end of the section and are preceded by a clear understanding of the relevant literature in relation to the ungulate behaviour and the theories relating to their productivity.

Materials and Methods

Study site and condition is thoroughly described. Reference is made to the use of MODIS to provide average snow cover conditions over a 15 year period. A valid approach given the almost daily temporal resolution of the system. Spatial resolution limitations should not be an issue. Reference should be made to how snow is identified in terms of spectral approach.

The survey unit selection section should comment on the MODIS resolution cell size for the used snow disappearance product. There is no issue with the use of this product but it could be described in more detail. Is it a 1km product?

Animal counts are appropriately conducted and logged for the study period. Appropriate justification is provided for the use of “ewe like” classification with reference to the variability associated with this assertion.

The use of SnowModel and its packages could be better justified with reference to the accuracy of the model. Reference to the environmental scenarios in which it has been tested and validated would be welcome. More consideration of its appropriate application to different latitude settings would also be appropriate here as would information about its handling of the continental location. Is there potential for overfitting the model with the large amount of inputs to be replicated for a particular site? How sensitive is the model and can it capture the variation exhibited in previous years accurately?

Justified assumptions are presented regarding landcover variation over the duration of the study period. Use of ASTER GDEM is validated but is a 60m product being used (2 arc second)? Does this study precede ASTER GDEM V3 where 1 arcsecond data is available?

Good evidence and methodology is provided regarding the calibration model to the site specific scenario using in situ data collections. More emphasis on the RMSE values would be welcome here to instil further confidence in the modelling approach.

A key improvement to make which will enhance the study is that the modelling approach could be much better visualised by using a work flow diagram. This would outline how each package interacts with one another and the external sources of calibrating data. It’s difficult to follow using the provided descriptive words alone particularly given the multi faceted aspect of the model.

Statistical analysis appears to be performed appropriately as described.

Results

Results are described in an adequate way. Statistical outputs are appropriately interpreted. Graphs are suitably presented.

I would recommend also reporting the RMSE values in relation to a mean value in the text to further establish the validity of the results. I assume that the RMSE values should be reported in metres to match with the data shown in Figure A3 and not the data displayed in the subsequent table (slight consistency error but changes the results quite significantly).

I'd like to see more reference to p values and significance regarding modelling results where possible.

Discussion

A question that remains. The model is calibrated using in situ data collection for calibration. Could the in situ measurements be replicated for a single year but not representative of long term variability or inter-site values? Is the remote camera data suitable for this purpose? Is there a case for overfitting here? Was the optimum setting tested independently to confirm its validity or was it only for snow depth? What are the problems arising from this?

An alternative explanation for snow’s effect on sheep productivity, where snow conditions in spring influence the vulnerability of lambs to predation, is harder to establish. How was this going to be established through this study? Should it be included as a hypothesis to test if there is very little evidence to justify this? Possibly better to consider this only in the discussion rather than setting the study out to address this.

There is more focus on general subject discussion rather than interpretation and critical analysis of the presented results. The section could be viewed as overly long as a consequence of this and seems to skirt around the key issues of the paper at times. A much more succinct format could be presented.

A much larger discussion should investigate the limitations of this particular modelling approach particularly with regards to calibrating using a single year of data in a long time series analysis.

Conclusion

Th modelling exercise has been conducted competently and has presented a very interesting finding of how snow condition and timing can impact Dall Sheep productivity.

A slight issue is that as a reader I am left wondering whether it is simply the temperature that is influencing the Lamb to Ewe ratio. The stated importance of season long environmental conditions would suggest this is a possibility. I would suggest addressing this factor as an appropriate further correction to make.

6. PLOS authors have the option to publish the peer review history of their article (what does this mean?). If published, this will include your full peer review and any attached files.

Reviewer #1: Yes: LUCA CORLATTI

Reviewer #2: Yes: Benjamin Larue

Reviewer #3: No

---

## [Author Response · Author response to Decision Letter 0]

12 Sep 2020

Reviewer #1: 

General comments:

My critical review of Cosgrove et al.’s paper on Seasonal influence of snow conditions on Dall’s sheep productivity will be fairly short, as ~70% of the paper (at last in the methods section) deals with subjects (i.e. remote sensing, climatic analyses) that are out of my area of expertise (wildlife/ungulate biology). I would thus recommend the editor to base his evaluation also on the comments of experts in these other fields.

That said, the MS appears generally well written, with mostly appropriate analyses (but see comments below) and suitable for the readership of PLoS ONE. After some adjustments, I think it will represent a nice addition to the existing literature on mountain ungulate ecology.

Specific comments:

l. 116-128: before these paragraphs you have correctly discussed several snow characteristics that can have an impact on sheep (and wildlife in general), but then here you use the somewhat fuzzy term ‘snow conditions’. What does that mean? Hardness? Persistence? Depth?

This has been altered now to read snow properties, which we hope is a more explicit term that links to the properties (depth/density) described in the previous paragraph

l. 190-195: so, the sheep counts were not spatially explicit, or? I am not sure I understand at which spatial scale you coupled counts and snow cover conditions.

An additional sentence has been included to clarify this. Only full surveys, where the entire Survey Unit was covered, are included in our dataset.

l. 195-204: I am not sure I understand here. So, lambs have been clearly identified I suppose, and ‘ewe-like’ counts have been pooled together with ‘true’ ewe counts, even though they might have been yearlings (of either sex) and/or young rams, right? If I misunderstood, please clarify. If I understood correctly, I am wondering why you would need to include them in the denominator in the first place. Can’t you simply use the ‘true’ ewe counts as a more robust (less error-prone) proxy of lamb-to-ewe ratio? Non-mature individuals and rams do not contribute to lambing anyway, so why would you include them in the counts?

You understand correctly, but the difficulties of judging these animals, especially from a plane, often means that surveyors end up including young animals of either sex into a ‘ewe-like’ category and do not attempt to classify or report any animal as a definite ‘ewe’. Our raw survey data has very few instances of yearlings and young rams being reported and we would question accuracy even when known-ewe categories are used. The section in question has been updated for clarity.

l.349-369: The stats are all in all alright, but can be much improved. First, it is not clear to me why you would use total counts as a weight in your model. Did you have issue of heteroskedasticity? Second, and most importantly, using only snow metrics as predictors is rather simplistic. Mountain ungulate demography is well known to be affected by the synergistic effect of climate and density (see Jacobson et al. 2004 or Corlatti et al. 2019). So, at least I would have expected the inclusion of the interactive effect of snow metrics with a proxy of population density (which should include yearling, rams and ewes, but not lambs), possibly with a delayed effect. Just looking at your data in Table 1 I suspect (but I might be wrong) that some density-dependent effect may be present. Other climatic features might also play a role, temperature for instance. The main point here is that snow may surely be the single most important parameter driving sheep dynamics, yet you ideally need to provide some more support for that (i.e. you need to convince the reader that no other parameter is driving the lamb-to-ewe ratio). Third, I am not sure why you did not test for at least a 1-year delayed effect of snow: the lamb-ewe ratio may well depend on the fact that the mother was not fertilized in the first place, because of poor body conditions, which might depend on the snow conditions of the year before.

Following this comment, and those from other reviewers, we have updated our statistical analyses (see 313 to 374). The counts were originally used as a weighting to buffer against the effect of small survey sizes having a disproportionate effect in our results. However, after further testing, including for heteroskedasticity, we did not see this play out and hence removed the weighting variable. This had the additional advantage of allowing a measure of R2 to be reported – which was to be marginal and conditional if the random effect of Survey Unit was still found to be significant. But due to the weighting variable being removed the random effect of Survey Unit was no longer important so we could instead fit models by Ordinary Least Squares and report R-sq. and adjusted R-sq. values instead.

We additionally tested for density dependence on lamb-to-ewe ratios but found no relationship, which follows the findings of van de Kerk et al. 2018, so did not include this in our model terms. Likewise, we included variables pertaining to a 1-year delayed effect of snow but did not find any of them to be a better predictor than the null model. Conversely, air temperature, which is an optional output from the MicroMet sub-model of SnowModel, was found to have some power as a predictor.

In the discussion we further include a critique of our focus largely on snow properties.

l. 396-407: a test and a measure of goodness of fit of your selected linear model are missing. Did you inspect model residuals? Can you provide a measure of R2 (marginal and conditional)?

In absence of clarifications with respect to potentially different drivers (i.e. density dependence, delayed effects on lamb-to-ewe ratio) is difficult to evaluate the Discussion.

See previous comment.

References

Jacobson, A.R., Provenzale, A., von Hardenberg, A., Bassano, B. & Festa-Bianchet, M. (2004). Climate forcing and density dependence in a mountain ungulate population. Ecology 85, 1598–1610.

Corlatti, L., Bonardi, A., Bragalanti, N. and L. Pedrotti. 2019. Long-term dynamics of Alpine ungulates suggest interspecific competition. Journal of Zoology 309, 241–249.

Reviewer #2: 

Comments to the Author

This manuscript explores how different snow properties affect the population dynamics of a northern alpine ungulate. More precisely, it explores the effects of snow depth, snow density, forageable area and snowfall on lamb to ewe ratios in Dall sheep. The authors use a spatially- explicit snow evolution model to determine snow cover properties. Subsequently, they evaluate the correlation between these properties and lamb to ewe ratios obtained from periodic summer aerial surveys which spanned over 37 years.

Overall, the manuscript is well structured and written. However, I think it could be slightly shortened by cutting certain sentences and avoiding excessive repetition of the same information in different sections. The analytical approach is elegant and convincing. However, I have never used nor read about spatially-explicit snow evolution models for my own research and I am not an expert on this specific matter. The research question isn’t so novel, but the methods to address it are. The results from this manuscript are hence novel and will be, in my opinion, of interest for management and conservation of ungulates and other species living in snowy environments. I enjoyed reviewing this manuscript and mostly have minor comments except for the advised use of sequential regression to confront multicollinearity and the over-interpretation of the AICc model selection. Fitting models with multiple snow properties and their interactions as covariates could greatly improve explanatory power and give a more accurate picture of the total effect of snow properties on Dall’s sheep productivity. I also think this manuscript could be improved and broadened be making it less “Dall sheep centric”. The application of SnowModel could also be interesting for studies of many plant and animal species living in snowy environments.

Introduction

The introduction could be slightly shortened by using less examples, more concise writing and less repetition. For example, at lines 116-117, the method used to determine the lamb to ewe ratios (aerial surveys) doesn’t need to be mentioned here and is simply a repetition of the what should remain in the methods.

The introduction has been edited and shortened by ~100 words

Line 42: I don’t think the reference to global climate systems is relevant to this manuscript.

Agreed and removed

Lines 55-61: Despite mentioning muskoxen, there is no citation pertaining to the species but there are 2 on deer. I think a citation on muskoxen is warranted.

The appropriate citation, which got lost in revisions prior to submission, has been put back in

Lines 86-88: I understand the idea here, but are there examples of studies where point-locations were non-representative? Though it is only one example, at my study site, which is also in an alpine environment, temperature is highly correlated (>0.85) to that of the nearest meteorological station in a valley-bottom ~15km away.

A highly cited paper (see below) exploring this is now referenced in the manuscript. It’s certainly true that temperature at valley bottom and alpine sites are likely to be highly correlated but snow properties are highly variable across spatial scales. In relatively dry, cold mountain environments like our study site the snowpack at the valley bottom is likely to accumulate without much disturbance from wind processes however in alpine areas over a distance of a few meters there can be snow-free areas and >2m deep drift features.

Molotch, N.P. and Bales, R.C., 2005. Scaling snow observations from the point to the grid element: Implications for observation network design. Water Resources Research, 41(11).

Lines 98-101: “These fluctuations are thought to be largely governed by variations in the production and survival of lambs” – The authors should also cite relevant articles that have highlighted the greater importance of juvenile survival and recruitment in ungulate population dynamics such as Gaillard et al. 1998 (Population dynamics of large herbivores: variable recruitment with constant adult survival; https://doi.org/10.1016/S0169-5347(97)01237-8 ) or Gaillard et al. 2000 (Temporal Variation in Fitness Components and Population Dynamics of Large Herbivores; https://doi.org/10.1146/annurev.ecolsys.31.1.367).

Gaillard et al. 1998 is now referenced

Lines 108-109: Citation on forage accessibility? References 79 or 80 from this manuscript?

Robinson et al. 2012 is now referenced

Lines 122-128: Very interesting contrasting hypotheses, but I don’t think the word predict should be used here. This study doesn’t exactly assess predictive power/accuracy using recognised methods such as cross-validation. In my opinion, a rephrasing like “...snow conditions established in the fall months should explain more variance in summer lamb-to-ewe ratios.” would be more appropriate.

Agreed – explain used now instead of predict

Methods

Line 142-143: Are average temperatures essential?

We believe that they are a useful indicator of the study area’s environment and hence likely snow regime/classification.

Lines 143-147: The description of lower elevation vegetation is not necessary and lengthy. It is indicated at line 151 that sheep range starts at around 1400 m. Also, it is later indicated that only pixels above 1200 m were selected at line 319.

Removed

Lines 194-195: Aerial surveys do not offer perfect detection. This could be especially true in Dall sheep which have a fission-fusion aggregation dynamic in which group size and composition can drastically change over the year. Potential biases of aerial surveys and the minimum count method have already been studied in Dall sheep, e.g. Udevitz et al. 2010 (Evaluation of Aerial Survey Methods for Dall's Sheep; https://doi.org/10.2193/0091- 7648(2006)34[732:EOASMF]2.0.CO;2 ) and Schmidt et al. 2012 (Using distance sampling and hierarchical models to improve estimates of Dall's sheep abundance; https://doi.org/10.1002/jwmg.216 ). Schmidt et al (2012) even highlight a bias in lamb abundance estimates which could be a serious issue in this manuscript. A sentence relating to the potential biases of these methods should be added.

We have address the above with the inclusion of the Schmidt reference, note of the known issues with the minimum count method, and further clarification that we only used full minimum count surveys in our dataset

Lines 206-209: I think this table should be in an appendix. It is lengthy and a sentence or two with descriptive statistics in the text (e.g. min and max date of flight, min and max number of individuals reported...) could replace it.

As suggested, the table has been moved to the appendix and a short summary of descriptive statistics is included instead 

Lines 210-346: I am not familiar with the specifics of SnowModel and its calibration, nor am I an expert on snow properties. I thus cannot comment these specific aspects.

Line 307: Is this not Root Mean Squared Error?

Corrected

Lines 318-333: This section could be condensed into only 3-4 short sentences by greatly cutting down on the descriptions of the cited studies (e.g. only including the citation is enough for the choice of altitudinal range).

Section has been shortened as suggested

Line 343: How high is half-chest height?

This is now stated in the manuscript (0.25 cm)

Lines 345-346: I would add “...Dall’s sheep productivity...” to avoid confusion with plant productivity because this part refers to forage.

Implemented

Lines 349-350: The developers of the R software should be cited. Use citation() in R to obtain the appropriate citation (https://astrostatistics.psu.edu/su07/R/library/utils/html/citation.html)

Implemented

Lines 350-352: The authors should consider including the date of the survey in their models given the relatively wide timespan in survey date (26 June to 4 August) and potential lamb summer mortality. I doubt this will have a large effect, but it is worth evaluating.

This was considered but previous work has not found survey date to effect lamb-to-ewe ratios – see van de Kerk et al. 2018 Appendix 2. We have included mention of this now in the Discussion section

Lines 354-355: Rather than including only one snow property variable per model, all four variables, a combination of variables and/or a combination of variables and their interactions should be tested using a sequential regression method. This exercise could be very informative if, as I expect (given lines 404-407), the full model or a model with a combination of different snow properties and their interactions is the most parsimonious and best explains variance in Dall’s sheep productivity. Sequential regression is done by including what is judged as the most important variable and then adding the residuals of the regression of a less important variable against the more important one. These residuals represent the effect of the second variable independent of the first. This process can be extrapolated to multiple collinear variables. See Graham 2003 (Confronting multicollinearity in ecological multiple regression; https://doi.org/10.1890/02-3114) and Dormann et al. 2013 (Collinearity: a review of methods to deal with it and a simulation study evaluating their performance; https://doi.org/10.1111/j.1600- 0587.2012.07348.x) for further explanations of the method.

Please see the reply above to Reviewer #1’s comment on our statistical analyses and the updated section in the manuscript.

 Results

Lines 389-394: The table or the figure should go in an appendix because they are mostly a repetition of the same information. I personally think the figure should stay.

Although the selected model was that of autumn snow depth, I think it would be interesting to have a figure showing the effect of the other more important snow properties. A 3 or 4 panel plot would be great.

Table has been moved to the appendices leaving a revised figure of the updated top model, we are reluctant to clutter the manuscript with further figures of other highly ranked models but welcome thoughts on this matter.

Discussion

Line 442: Remove “an iconic mountain ungulate”.

Removed

Lines 442-444: “...conditions that inhibit Dall’s sheep, ...” Dall sheep by themselves can’t be inhibited. Their productivity or recruitment, however, can. I think what is meant here is simply: “Snow conditions, most notably increased snow depth in the fall, were strongly associated with declines in Dall’s sheep productivity”.

Adjusted following suggestion

Lines 457-461: Distinguishing between effects of predation and foraging efficiency/forage quantity seems far stretched given the results in this manuscript. Spring snow conditions can also influence productivity because of their influence on the length of the growing season. However, I agree that snow conditions in fall appear more important than snow conditions in spring for Dall’s sheep productivity.

This section has now been revised

Line 460: Remove “that”.

As above

Line 470: I am not sure I understand this. Shouldn’t it be “...immediately after lambing.”.

Adjusted to read ‘…immediately before or after lambing.’

Lines 486-492: Care must be taken when interpreting an AICc model selection. The fact that Fall forageable area was the third highest ranked model is not indicative that the “...methodology had some power in mapping the extent of snow conditions that could potentially impact foraging.”. AICc values are in no way representative of power nor do they represent the proportion of the variance explained. For instance, a model could be the highest ranked model from a candidate set based on an AICc model comparison and still explain <1% of the variance if the other modelsare even worst. Instead, to evaluate and compare model performance, I suggest using the rsquared function from the piecewiseSEM package in R (https://rdrr.io/cran/piecewiseSEM/man/rsquared.html). This function returns (pseudo)-R2 values for all generalized linear mixed effects models.

This section has been revised in line with the new statistical analyses and R2 values are now reported and discussed.

Reviewer #3: 

This is a very relevant modelling study conducted to link ungulate productivity with environmental constraints on the landscape, particularly in relation to snow cover and other associated variables. Use is made of remote sensing data to establish snow cover extent and timings and to verify topographical distribution of key study areas. Use is made of a long time series of Dall Sheep population data to establish relationships between these population numbers, particularly ewe to lamb ratios, and modelled snow cover over the duration of the study using a year of environmental data to constrain the model. The suitability of the model is determined through RMSE analysis relating to the in situ data with snow depth from remote cameras used as the independent verification source.

Abstract

The study is well outlined in the abstract. There is clear emphasis of the research question to be addressed as well as providing some descriptions of the results. Quantitative reporting of the regression and statistical key results would be welcome here in more depth. Another thing to consider in the abstract is to report the RMSE of the model to the snow characteristics you are trying to replicate. This is an essential part of the study. Additionally it would be useful to discount the role of temperature in these relationships at this stage so that the focus is clearly established on snow cover independent of temperature.

RMSE and bias of the modelled snow data to the remote cameras has now been included. Likewise model statistics and description of the top-ranked model that includes fall temperature as a term.

Introduction

The hypotheses of the study are well presented at the end of the section and are preceded by a clear understanding of the relevant literature in relation to the ungulate behaviour and the theories relating to their productivity.

Materials and Methods

Study site and condition is thoroughly described. Reference is made to the use of MODIS to provide average snow cover conditions over a 15 year period. A valid approach given the almost daily temporal resolution of the system. Spatial resolution limitations should not be an issue. Reference should be made to how snow is identified in terms of spectral approach.

The survey unit selection section should comment on the MODIS resolution cell size for the used snow disappearance product. There is no issue with the use of this product but it could be described in more detail. Is it a 1km product?

Updated to include reference to 500 m resolution

Animal counts are appropriately conducted and logged for the study period. Appropriate justification is provided for the use of “ewe like” classification with reference to the variability associated with this assertion.

The use of SnowModel and its packages could be better justified with reference to the accuracy of the model. Reference to the environmental scenarios in which it has been tested and validated would be welcome. More consideration of its appropriate application to different latitude settings would also be appropriate here as would information about its handling of the continental location. Is there potential for overfitting the model with the large amount of inputs to be replicated for a particular site? How sensitive is the model and can it capture the variation exhibited in previous years accurately?

References to SnowModel’s previous use in Alaska have now been included. The references for each of SnowModel’s submodels include descriptions of their physics and validation and a note now added after them indicating this.

Some discussion of the modelling uncertainties is included at the end of the Discussion previously that address the above concerns towards overfitting and model sensitivity. With a limited calibration/validation dataset for the field site it is hard to comprehensively assess model performance but SnowModel is highly regarded and much used by the snow science community in a diverse range of settings and applications.

Justified assumptions are presented regarding landcover variation over the duration of the study period. Use of ASTER GDEM is validated but is a 60m product being used (2 arc second)? Does this study precede ASTER GDEM V3 where 1 arcsecond data is available?

A 1 arc second product is being used as referenced on line 239 in the latest ms.

Good evidence and methodology is provided regarding the calibration model to the site specific scenario using in situ data collections. More emphasis on the RMSE values would be welcome here to instil further confidence in the modelling approach.

This suggestion is slightly confusing – do you mean the RMSE values reported in the SnowModel calibration of the results section?

A key improvement to make which will enhance the study is that the modelling approach could be much better visualised by using a work flow diagram. This would outline how each package interacts with one another and the external sources of calibrating data. It’s difficult to follow using the provided descriptive words alone particularly given the multi faceted aspect of the model.

A workflow diagram of the modelling procedure has been included in the supplementary materials. To produce this with a visual description of the calibration/validation/assimilation procedures produced an overly complex diagram so a simple timestep to timestep depiction is used.

Statistical analysis appears to be performed appropriately as described.

Results

Results are described in an adequate way. Statistical outputs are appropriately interpreted. Graphs are suitably presented.

I would recommend also reporting the RMSE values in relation to a mean value in the text to further establish the validity of the results. I assume that the RMSE values should be reported in metres to match with the data shown in Figure A3 and not the data displayed in the subsequent table (slight consistency error but changes the results quite significantly).

I'd like to see more reference to p values and significance regarding modelling results where possible.

As with a previous comment I am unsure as to what the reviewer means when mentioning RMSE in relation to the mean value here? RMSE has been updated to metres to match with the table in the appendix – an oversight in the original manuscript.

Significance is referred to for each term in the highest ranked model.

Discussion

A question that remains. The model is calibrated using in situ data collection for calibration. Could the in situ measurements be replicated for a single year but not representative of long term variability or inter-site values? Is the remote camera data suitable for this purpose? Is there a case for overfitting here? Was the optimum setting tested independently to confirm its validity or was it only for snow depth? What are the problems arising from this?

This is an important point which is alluded to in the final paragraph of the discussion in the original manuscript and has now been extended upon. It is quite possible for the model to be overfitted for the only year where we have data to calibrate/validate to. However, given the coarseness generated by aggregating the snow covariates in time (seasonally) and space (across the modelling domain) it is likely that the model captures the magnitude of the interannual variability of snow properties reasonably well as it relies on well-proven physics to do so. 

An alternative explanation for snow’s effect on sheep productivity, where snow conditions in spring influence the vulnerability of lambs to predation, is harder to establish. How was this going to be established through this study? Should it be included as a hypothesis to test if there is very little evidence to justify this? Possibly better to consider this only in the discussion rather than setting the study out to address this.

This has now been removed from the hypotheses.

There is more focus on general subject discussion rather than interpretation and critical analysis of the presented results. The section could be viewed as overly long as a consequence of this and seems to skirt around the key issues of the paper at times. A much more succinct format could be presented.

A much larger discussion should investigate the limitations of this particular modelling approach particularly with regards to calibrating using a single year of data in a long time series analysis.

In response to this comment and those of the other reviewers the discussion has been reworked and, hopefully, refined to include more critical analysis of our results and methodology while maintaining its original length. We feel that a larger discussion investigating the limitations of the modelling approach in depth would confuse the intention and subject of the paper. Difficulties with modelling snow in complex environments are well known and we make an effort to contrast the relative merits and disadavantages of using a modelling approach vs remote sensing in settings with little in-situ data

Conclusion

Th modelling exercise has been conducted competently and has presented a very interesting finding of how snow condition and timing can impact Dall Sheep productivity.

A slight issue is that as a reader I am left wondering whether it is simply the temperature that is influencing the Lamb to Ewe ratio. The stated importance of season long environmental conditions would suggest this is a possibility. I would suggest addressing this factor as an appropriate further correction to make.

We hope the updated statistical analyses and the inclusion of air temperature as a covariate addresses this issue.

---

## [Decision Letter · Decision Letter 1]

4 Nov 2020

PONE-D-20-03620R1

Seasonal influence of snow conditions on Dall’s sheep productivity in Wrangell-St Elias National Park and Preserve

PLOS ONE

Dear Dr. Cosgrove,

Thank you for submitting your manuscript to PLOS ONE. After careful consideration, we feel that it has merit but does not fully meet PLOS ONE’s publication criteria as it currently stands. Therefore, we invite you to submit a revised version of the manuscript that addresses the points raised during the review process. The reviewer #3, has just contact me to include the need of disscussing the low R squared values of your results with respect to the results of other studies that they are contradicting.

We look forward to receiving your revised manuscript.

Kind regards,

Emmanuel Serrano, PhD

Academic Editor

PLOS ONE

Reviewers' comments:

Reviewer's Responses to Questions

**Comments to the Author**

1. If the authors have adequately addressed your comments raised in a previous round of review and you feel that this manuscript is now acceptable for publication, you may indicate that here to bypass the “Comments to the Author” section, enter your conflict of interest statement in the “Confidential to Editor” section, and submit your "Accept" recommendation.

Reviewer #1: All comments have been addressed

Reviewer #2: All comments have been addressed

Reviewer #3: All comments have been addressed

2. Is the manuscript technically sound, and do the data support the conclusions?

Reviewer #1: Yes

Reviewer #2: Yes

Reviewer #3: Yes

3. Has the statistical analysis been performed appropriately and rigorously? 

Reviewer #1: Yes

Reviewer #2: Yes

Reviewer #3: Yes

4. Have the authors made all data underlying the findings in their manuscript fully available?

Reviewer #1: Yes

Reviewer #2: Yes

Reviewer #3: Yes

5. Is the manuscript presented in an intelligible fashion and written in standard English?

Reviewer #1: Yes

Reviewer #2: Yes

Reviewer #3: Yes

6. Review Comments to the Author

Reviewer #1: I thank the author for their detailed answer to my queries. I think the paper is a nice contribution to the existing literature on ungulate population dynamics, and I would recommend publication, after some very minor adjustments (see below).

I appreciate that the authors tested the effect of density dependence and of other climatic variables with delayed effects in their models. While I do understand that it makes sense to exclude density dependence and delayed effects from the final models, I would still recommend to at least mention that preliminary analyses revealed no density dependence and no delayed effects. Density dependence is a major driver of ungulate population dynamics (see references provided in the first round of review), and I think the reader needs to be reassured that you tested for this effect, even if only at a preliminary stage. One or 2 sentences would suffice.

I still cast doubts about the choice of pooling young rams and young ewes in a ewe-like category, but I do not know the field situation there, and the authors provide sufficient information to allow the reader understand and evaluate their procedure, so I am fine with it. Perhaps, instead of lamb-to-ewe, I would use “birth rate” or something the like, this would likely create less confusion in the reader.

l. 123-134: I don’t quite get why in their hypotheses the authors did not include the winter effect on lamb-to-ewe ratio. Clearly, the effect hypothesized in H1 can owe to winter conditions, so why would they only mention the fall months? (also considering that winter months were included in the list of models).

Best wishes,

Luca Corlatti

Reviewer #2: My comments have adequately been addressed. The manuscript has been shortened were it should and the statistical analyses have been greatly improved. Limitations to the study have also been addressed.

Reviewer #3: This is a very much improved manuscript. The changes that have been made are quite significant and follow the guidance of the reviewers very closely. From my perspective my concerns have largely been addressed either through corrections or justifications and their remains only a few typographical errors that remain to be corrected. A slight issue in the review process was the inability to see the tabulated data fully in the new submission via the track changes, as such I was unable to immediately confirm the reported values in Table 1.

With respect to the author reply, the comment regarding RMSE values in the earlier section was suggested for inclusion to outline the performance of the calibration regarding snow depth and bulk snow density. Being part of the calibration process rather than a result of the study, my suggestion was to include the RMSE of the calibration at this earlier methodological stage. This is a matter of choice, and I'm happy for you to report both of these results at the start of the results section if you choose to do so. Reporting the bulk snow density RMSE would be appropriate alongside the snow depth RMSE declaration in the results section if you decide on this approach.

A particular typo to be aware of is the reference to 30 years of data early in the manuscript (Line 92 in track changes document) and later referred to as 37 years at Line 519. Further context to the statement of "with less predictive power" could also be provided on Line 524. Very minor issues alongside the other typos that can be corrected through further proof reading.

Accept following the addressing of these very minor concerns. An enjoyable study to read about.

7. PLOS authors have the option to publish the peer review history of their article (what does this mean?). If published, this will include your full peer review and any attached files.

Reviewer #1: **Yes: **Luca Corlatti

Reviewer #2: **Yes: **Benjamin Larue

Reviewer #3: **Yes: **Dr Matthew Brolly

---

## [Author Response · Author response to Decision Letter 1]

16 Dec 2020

Response to the review PONE-D-20-03620 ‘Seasonal influence of snow conditions on Dall’s sheep productivity in Wrangell-St Elias National Park and Preserve’

Dear Dr Serrano,

We are grateful to you and the reviewers for the further comments on our manuscript and are glad that our original revisions were well received. As the comments were relatively few, I will address them individually in the following paragraphs.

In your emailed response you noted that Reviewer 3 was concerned with the R-squared values we report relative to other cited studies. To address this, I have included a comparison on lines 499 to 502 of our top-model to the R-squared reported by Rattenbury et al. (2018) for a region that is within our study area, Nabesna. We report an adjusted R-squared of 0.41, whereas Rattenbury et al. (2018), using the remote-sensing derived date of the end of the continuous snow season as their predictor of lamb-to-ewe ratios, report an R-squared of 0.33 (see figure 4 in Rattenbury et al.). The similar study of van de Kerk et al. (2018) only reported AICc for their various models, hence direct comparison is not possible. Of note is that Rattenbury et al. (2018) report a combined R-squared of 0.31 for all five Dall’s sheep count areas included in their study, and an R-squared of 0.65 for the Itkillik sub-area alone, which is the only sub-area to have an R-squared > 0.41. However, I believe it’s not appropriate to compare results from quite different Dall’s sheep ranges, Itkillik, for example, is in the Brooks Range where the snow season and landscape is dissimilar to the Wrangells, so only include Rattenbury et al.’s (2018) Nabesna result in the text. In response to why our R-squared could be perhaps thought of as relatively low, we offer the existing discussion between lines 539 and 552, which discusses other factors affecting Dall’s sheep productivity.

Reviewer 1 asked for the preliminary analysis of density dependence and delayed effects to be mentioned within the text. We appreciate this point and I have cleared up the discussion of this on lines 521 to 527 to be more explicit as to what we did to address in terms of density dependence. Likewise, I have hopefully clarified our delayed effect results (see line 407) and guide the reviewer to consider lines 534 to 538, where this result is reflected upon in the Discussion section. A further comment considered our choice of using ‘lamb-to-ewe’ to describe our metric of productivity and suggested using different terminology. We are sympathetic to the reviewer’s point, especially in respect to the potential inclusion of young-rams into the ‘ewe’ pool. However, we do feel confident that our explanation of the metric, and how it was derived, is clear with careful reading (see lines 192 to 204) and that it is appropriately and consistently termed in respect to similar, contemporary studies, e.g. van de Kerk et al. (2018) and Rattenbury et al. (2018). Reviewer 1 also makes a salient point on our hypotheses, questioning why we also analyse winter snow and climate conditions when our hypotheses only address fall and spring. This was an oversight on our part, so we have updated hypothesis 1 (lines 120 to 123) to now include; ‘…in which case snow conditions established in the fall months and persisting through the winter months should better explain summer lamb-to-ewe ratios’. The original idea in including the winter months in the analysis was to capture persistent effects of snow cover in the instance of a relatively mild and low-snow early fall followed by a relatively ‘snowy’ and cold late-fall and winter, which might still show a persistent effect on productivity vs spring conditions alone. 

Reviewer 3 additionally discussed the location within the text of the results from the SnowModel calibration and the reporting of the bulk density RMSE in them. Prior to submission we considered whether the SnowModel calibration results should be contained within the Methods section but came to the opinion that, as they were results in and of themselves, they were more appropriately placed in the Results section. We also felt that it was possibly more readable in this format. The bulk density RMSE is now reported on line 382. Lastly, Reviewer 3 mentioned some typographical errors – after further proofreading we hope that we have caught these, especially those relating to incorrectly numbered references due to inadequate use of my referencing software.

We hope that you find the above and the updated manuscript to your satisfaction, and thank you again for the time taken to review it with care. We look forward to your response.

Sincerely,

Chris Cosgrove

---

## [Editor Report · Decision Letter 2]

17 Dec 2020

Seasonal influence of snow conditions on Dall’s sheep productivity in Wrangell-St Elias National Park and Preserve

PONE-D-20-03620R2

Dear Dr. Cosgrove,

We’re pleased to inform you that your manuscript has been judged scientifically suitable for publication and will be formally accepted for publication once it meets all outstanding technical requirements.

Kind regards,

Emmanuel Serrano, PhD

Academic Editor

PLOS ONE

Additional Editor Comments (optional):

Congratulations!

I hope you will have a merry Christmas and happy new year

Emmanuel
---

## [Editor Report · Acceptance letter]

22 Dec 2020

PONE-D-20-03620R2 

Seasonal influence of snow conditions on Dall’s sheep productivity in Wrangell-St Elias National Park and Preserve 

Dear Dr. Cosgrove:

I'm pleased to inform you that your manuscript has been deemed suitable for publication in PLOS ONE. Congratulations! Your manuscript is now with our production department. 

Kind regards, 

on behalf of

Dr. Emmanuel Serrano 

Academic Editor

PLOS ONE